

# Comparative masticatory myology in anteaters and its implications for interpreting morphological convergence in myrmecophagous placentals

Sérgio Ferreira-Cardoso[1], Pierre-Henri Fabre[1,2], Benoit de Thoisy[3,4], Frédéric Delsuc[1] and Lionel Hautier[1,2]

[1] CNRS, IRD, EPHE, Université de Montpellier, Institut des Sciences de l'Evolution de Montpellier (ISEM), Montpellier, France
[2] Mammal Section, Life Sciences, Vertebrate Division, The Natural History Museum, London, United Kingdom
[3] Institut Pasteur de la Guyane, Cayenne, French Guiana, France
[4] Kwata NGO, Cayenne, French Guiana, France

Corresponding authors
Sérgio Ferreira-Cardoso, sergio.ferreira-cardoso@umontpellier.fr
Lionel Hautier, lionel.hautier@umontpellier.fr

## ABSTRACT

**Background**. Ecological adaptations of mammals are reflected in the morphological diversity of their feeding apparatus, which includes differences in tooth crown morphologies, variation in snout size, or changes in muscles of the feeding apparatus. The adaptability of their feeding apparatus allowed them to optimize resource exploitation in a wide range of habitats. The combination of computer-assisted X-ray microtomography ($\mu$-CT) with contrast-enhancing staining protocols has bolstered the reconstruction of three-dimensional (3D) models of muscles. This new approach allows for accurate descriptions of muscular anatomy, as well as the quick measurement of muscle volumes and fiber orientation. Ant- and termite-eating (myrmecophagy) represents a case of extreme feeding specialization, which is usually accompanied by tooth reduction or complete tooth loss, snout elongation, acquisition of a long vermiform tongue, and loss of the zygomatic arch. Many of these traits evolved independently in distantly-related mammalian lineages. Previous reports on South American anteaters (Vermilingua) have shown major changes in the masticatory, intermandibular, and lingual muscular apparatus. These changes have been related to a functional shift in the role of upper and lower jaws in the evolutionary context of their complete loss of teeth and masticatory ability.

**Methods**. We used an iodine staining solution ($I_2KI$) to perform contrast-enhanced $\mu$-CT scanning on heads of the pygmy (*Cyclopes didactylus*), collared (*Tamandua tetradactyla*) and giant (*Myrmecophaga tridactyla*) anteaters. We reconstructed the musculature of the feeding apparatus of the three extant anteater genera using 3D reconstructions complemented with classical dissections of the specimens. We performed a description of the musculature of the feeding apparatus in the two morphologically divergent vermilinguan families (Myrmecophagidae and Cyclopedidae) and compared it to the association of morphological features found in other myrmecophagous placentals.

**Results**. We found that pygmy anteaters (*Cyclopes*) present a relatively larger and architecturally complex temporal musculature than that of collared (*Tamandua*) and giant (*Myrmecophaga*) anteaters, but shows a reduced masseter musculature, including

the loss of the deep masseter. The loss of this muscle concurs with the loss of the jugal bone in Cyclopedidae. We show that anteaters, pangolins, and aardvarks present distinct anatomies despite morphological and ecological convergences.

## INTRODUCTION

The Cretaceous terrestrial revolution and the Cretaceous-Paleogene (K-Pg) mass extinction event are often viewed as milestones in placental mammal evolution (*Meredith et al., 2011*). These events promoted the opening of terrestrial ecological niches available to placentals, contributing to their morphological diversification (*Romer, 1974*; *Alroy, 1999*; *Halliday et al., 2019*). The adaptation of the placental feeding apparatus likely contributed to this radiation (*Price et al., 2012*), as mechanical processing of food items is essential to ensure a better energetic intake (*Hiiemae, 2000*). In mammals, food processing essentially occurs via mastication, which mainly consists of mandibular adduction/abduction (sagittal and coronal planes motion; e.g., *Herring & Scapino, 1973*). Mandibular elevation during adduction is performed by the temporal and the masseter, while transverse movements involve mainly the internal pterygoid (*Kendall et al., 1993*; *Hylander, 2006*). Despite the homogeneity of the main complexes of the masticatory apparatus among placental mammals, muscular architecture and proportions vary largely (e.g., *Parsons, 1896*; *Toldt, 1905*; *Turnbull, 1970*). This suggests a wide range of functional disparity associated with both phylogenetic constraints and ecological specialization (*Samuels, 2009*; *Hautier, Lebrun & Cox, 2012*; *Fabre et al., 2017*; *Ginot, Claude & Hautier, 2018*; *Kohli & Rowe, 2019*).

Among ecological factors, dietary specialization is often considered to be a major driver of cranial morphological specialization (*Varrela, 1990*; *Barlow, Jones & Barratt, 1997*; *Nogueira, Peracchi & Monteiro, 2009*; *Hautier, Lebrun & Cox, 2012*; *Klaczko, Sherratt & Setz, 2016*; *Maestri et al., 2016*). In placental mammals, the evolution of myrmecophagy (ant- and termite-eating) is a textbook example of morphological convergence driven by diet (*McGhee, 2011*). It evolved in three of the four major placental clades (e.g., *Springer et al., 2013*), including Laurasiatheria (pangolins and aardwolf), Afrotheria (aardvark), and Xenarthra (anteaters and giant armadillo). Morphological convergence associated with myrmecophagy is such that early classifications grouped pangolins, aardvarks, and xenarthrans in the monophyletic Edentata (toothless; *Vicq-d'Azyr, 1742*; *Cuvier, 1798*). In these taxa, convergent cranial traits related to ant- and termite-eating include tooth reduction or complete loss, extreme snout elongation, and long extensible tongues (*Rose & Emry, 1993*; *Davit-Béal, Tucker & Sire, 2009*; *Ferreira-Cardoso, Delsuc & Hautier, 2019*; *Gaudin et al., 2020*). Additionally, myrmecophagy led to the loss of the ability to chew in anteaters, pangolins, and giant armadillos (*Naples, 1999*; *Davit-Béal, Tucker & Sire, 2009*; *Vizcaíno et al., 2009*).

Anteaters are a good example of the morphofunctional adaptation to myrmecophagy, with their specialized feeding apparatus musculature (*Reiss, 1997*; *Naples, 1999*; *Endo et al., 2007*; *Endo et al., 2017*) associated with unique skeletal features such as edentulous jaws, unfused mandibular symphysis, and extremely reduced coronoid and angular processes (with the exception of *Cyclopes*). South American anteaters (Vermilingua, Xenarthra) consist of ten currently recognized extant species (*Reeve, 1940*; *Wetzel, 1985*; *Hayssen, 2011*; *Navarrete & Ortega, 2011*; *Hayssen, Miranda & Pasch, 2012*; *Gaudin, Hicks & Di Blanco, 2018*; *Miranda et al., 2018*) that split from their sloth sister-group around 58 million years ago (*Gibb et al., 2016*). The monogeneric Cyclopedidae comprises the pygmy anteaters (*Cyclopes* spp.), a group of small arboreal recently described cryptic species (*Miranda et al., 2018*) feeding solely on ants (*Montgomery, 1985*; *Redford, 1987*; *Hayssen, Miranda & Pasch, 2012*). The Myrmecophagidae include two ant- and termite-eating genera (*Montgomery, 1985*; *Hayssen, 2011*): the semi-arboreal collared anteater (*Tamandua tetradactyla*) and northern tamandua (*Tamandua mexicana*), and the terrestrial giant anteater (*Myrmecophaga tridactyla*; *Wetzel, 1985*; *Gaudin & Branham, 1998*; *Gibb et al., 2016*). Despite their similar diets and prey capture strategies (*Montgomery, 1983*; *Montgomery, 1985*), the extreme elongation of the myrmecophagid rostrum and the loss of the jugal bone in cyclopedids are illustrative examples of morphological differences between the two families. Moreover, the Cyclopedidae present several peculiar morphological features such as the strongly curved basicranial/basifacial axis or the plesiomorphic unfused pterygoid bones (*Gaudin & Branham, 1998*). Yet, the masticatory musculature of the putatively most diversified anteater family (*Miranda et al., 2018*) has only been briefly explored (*Reiss, 1997*). A comparative study of the muscles involved in mastication would be key to assess if the adaptation to myrmecophagy constrained the degree of morphofunctional disparity within the Vermilingua.

The unique morphology of anteaters has intrigued early anatomists. *Rapp (1852)* provided the first description of the myology of the collared anteater (*T. tetradactyla*), but did not include the head musculature. *Owen (1856)* and *Pouchet (1874)* described the limb and head muscles of the giant anteater. *Galton (1869)*, *Humphry (1869)*, and *Macalister (1875)* studied the myology of the pygmy anteater (*C. didactylus*), but once again did not consider the head muscles. More recently, *Naples (1985a)* provided a detailed description of the superficial musculature of the head for all pilosans, including the three anteater genera. However, *Reiss (1997)* was the first to provide a comprehensive description of the head musculature of the northern tamandua, using the pygmy and giant anteaters mostly for comparisons. *Naples (1999)* and *Endo et al. (2007)* provided a thorough description of the masticatory musculatures of the giant anteater. Both authors suggested that the reduced masticatory muscles reflect a functional shift from the typical adduction/abduction cycle towards a predominantly hemimandibular rotation about the anteroposterior axis (roll). *Endo et al. (2017)* described the masticatory muscles of the collared anteater, highlighting their similarities with those of the giant anteater. The studies listed above concurred on two main points: (i) the masticatory musculature is reduced in all anteaters, when compared to their sloth sister-group or to other placental mammals (*Naples, 1985b*; *Naples, 1999*),

and (ii) the modified hyolingual apparatus (protruding elongated tongue) coincides with a functional shift of the masticatory apparatus (roll-dominated mandibular movements).

Here, we describe the masticatory, facial-masticatory, and intermandibular muscles in the three anteater genera *Cyclopes*, *Tamandua*, and *Myrmecophaga* (*Gaudin & Branham, 1998*). We used a combination of traditional and virtual dissections to accurately measure muscular mass and volumes, while reconstructing 3D surfaces based on iodine-enhanced µCT-scanning (e.g., *Gignac & Kley, 2014*; *Ginot, Claude & Hautier, 2018*). Our study aims to provide the first comprehensive description of the masticatory apparatus of the three anteater genera. Finally, we compare our results to existing data from other myrmecophagous placentals (pangolins and the aardvark). We hypothesize that the convergent reduction/loss of mastication linked to myrmecophagy was accompanied with similar muscular morphologies.

## MATERIALS AND METHODS

### Biological sampling

We dissected specimens from the three extant anteater genera: *Cyclopes didactylus* ($n = 2$); *Tamandua tetradactyla* ($n = 3$); *Myrmecophaga tridactyla* ($n = 1$). *C. didactylus* specimens (M1525_JAG, M1571_JAG) and one specimen of *T. tetradactyla* (UM-778-N) were alcohol-preserved collection specimens previously fixed in a 10% formaldehyde solution. *T. tetradactyla* (M3074_JAG) and *M. tridactyla* (M3023_JAG) were frozen collection specimens. *T. tetradactyla* specimens correspond to wild roadkills while *M. tridactyla* was a zoo specimen (M3023_JAG). M3075_JAG (*T. tetradactyla*) was immediately dissected after collection along the road. All heads were extracted and, when possible, the complete sternum and the tongue musculature were also detached (M1525_JAG, M3075_JAG). Frozen and fresh heads were then fixed in a 10% formaldehyde solution to allow for long term storage. All specimens were stored in 70% ethanol. All wild specimens were collected in French Guiana and were stored in the collections of the Association Kwata (JAGUARS collection, Cayenne, France) and the Université de Montpellier (UM; Montpellier, France).

### Conventional dissections

For each specimen, only one side was dissected. The areas of insertion and origin were described and each muscle was then stored separately in a 70% ethanol solution. All muscles were posteriorly removed from the ethanol solution and weighted with a Sartorius A 120 S precision weighing scale (precision = 0.01 mg). Individual wet muscle masses are provided as Supplemental Tables. Muscular volume was calculated for the three stained specimens based on mass and a density of 1.06 g cm$^{-3}$ (*Murphy & Beardsley, 1974*). These estimations were then compared to the volumes obtained with the digital segmentations. All dissected specimens were re-stored in a 70% ethanol solution for a period no longer than two weeks, prior to staining (see below).

### Iodine-enhanced CT-scanning

For each species, the most complete and well-preserved specimen (Figs. S1A–S1C) was selected to be stained. Contrast-enhanced µCT-scans result in an increase of density of the

soft tissues and thus the contrast between muscles and bone is lost (*Cox & Jeffery, 2011*). Therefore, the specimens were μCT-scanned prior to staining, so that the bone tissue could be easily reconstructed. A second scan was performed after staining (see below). High-resolution microtomography (μCT) was performed at Montpellier Rio Imaging (MRI; Microtomograph RX EasyTom 150, X-ray source 40–150 kV) platform. Original voxel sizes were 35.0 μm for *C. didactylus* (M1571_JAG), 76.0 μm for *T. tetradactyla* (M3075_JAG), and 112.1 μm for *M. tridactyla* (M3023_JAG).

The contrast enhancement protocol was adapted from *Cox & Jeffery (2011)*. All specimens were removed from the 70% ethanol solution and directly transferred to a solution of iodine (5% $I_2KI$) for a period of two to eight weeks, depending on size. This concentration represents a trade-off between observed staining efficiency and the soft-tissue shrinkage associated with iodine staining, even if incubation period seems to have a limited effect in soft-tissue shrinkage, after the first two days (*Vickerton, Jarvis & Jeffery, 2013*). In *T. tetradactyla* and *M. tridactyla*, small volumes of $I_2KI$ solution were directly injected into the muscles, as the large size of the specimens hinders an efficient passive diffusion of the contrasting agent.

The contrast-enhanced scans were imported to Fiji (*Schindelin et al., 2012*) and 2-fold binning was performed in order to allow for a better handling of the three-dimensional (3D) volumes. 3D volumes of each muscle were generated using Avizo 9.7.0 (Thermo Fisher Scientific). We generated surfaces for the skull and muscles separately and then used the function "register" in Avizo 9.7.0 to align these reconstructions. Most tendons and aponeuroses were not stained by the iodine solution, and were therefore not reconstructed. Some muscles may thus appear artificially detached from the skull (e.g., *M. masseter superficialis* in myrmecophagids).

### Nomenclature

We used the muscular nomenclature for the masticatory apparatus of xenarthrans defined by *Naples (1985a)*, *Naples (1985b)* and *Naples (1999)*. More recent descriptions of the masticatory apparatuses of *M. tridactyla* and *T. tetradactyla* adopted the English version of the same terminology (*Endo et al., 2007*; *Endo et al., 2017*). All muscle names are fully written in Latin. We follow *Naples (1999)* in using the term '*pars*' to address myologically distinct units with developmentally common origins (e.g., *M. buccinatorius pars externa* vs *M. buccinatorius pars interna*), while the term '*pars reflexa*' is used here to characterize a part of a myological unit which wraps around a bone structure (e.g., *Cox & Jeffery, 2015*). Muscle abbreviations are provided in Table 1.

## RESULTS

Measurements of the muscles involved in mastication are summarized in Tables 2 and 3. Volume measurements were performed on the segmented muscles of the contrast-enhanced specimens. Mass measurements of all dissected specimens are provided in Table S1. Volumes estimated from muscle weights are correlated with those obtained from the 3D-reconstructions for the three specimens (all $p < 0.05$; Table S2). In *C. didactylus* and *T. tetradactyla* the volumes estimated from the mass were smaller than those

**Table 1  Abbreviations of the illustrated muscles.** This list includes masticatory, facial-masticatory, intermandibular, and hyoid muscles.

| Muscle | Abbreviation | Muscle | Abbreviation |
|---|---|---|---|
| *M. masseter profundus* | M.m.p. | *M. pterygoideus internus pars anterior* | pa-M.p.i. |
| *M. masseter superficialis* | M.m.s. | *M. pterygoideus internus pars posterior* | pp-M.p.i. |
| *M. masseter superficialis pars anterior* | pa-M.m.s. | *M. mandibuloauricularis* | M.ma. |
| *M. masseter superficialis pars posterior* | pp-M.m.s. | *M. buccinatorius pars externa* | pe-M.b. |
| *M. temporalis superficialis* | M.t.s. | *M. buccinatorius pars interna* | pi-M.b. |
| *M. temporalis superficialis pars zygomatica* | pz-M.t.s. | *M. mylohyoideus pars posterior* | pp-M.mh. |
| *M. temporalis profundus pars lateralis* | pl-M.t.p. | *M. intermandibularis anterior* | M.i.a. |
| *M. temporalis profundus pars medialis* | pm-M.t.p. | *M. mylohyoideus pars anterior* | pa-M.mh. |
| *M. pterygoideus externus pars superior* | ps-M.p.e. | *M. interstylohyoideus* | M.ish. |
| *M. pterygoideus externus pars inferior* | pi-M.p.e. | *M. geniohyoideus* | M.gh. |
| *M. pterygoideus internus* | M.p.i. | *M. mastostyloideus* | M.mst. |

**Table 2  Masticatory muscle volumes (mm³; left column) and percentages (right column) obtained from the 3D models of the contrast-enhanced specimens segmentation.**

| Muscles | \multicolumn Volume in mm³/Masticatory volume (%) | | | | | |
|---|---|---|---|---|---|---|
| | *C. didactylus* | | *T. tetradactyla* | | *M. tridactyla* | |
| **M.t.s.** | 139.8 | 42.2 | 575.2 | 20.9 | 1718.1 | 13.5 |
| **pz-M.t.s.** | 8.0 | 2.4 | 153.9 | 5.6 | 1198.0 | 9.4 |
| **pm-M.t.p.** | 19.5 | 5.9 | 120.2 | 4.4 | 510.2 | 4.0 |
| **pl-M.t.p.** | 33.6 | 10.1 | 128.0 | 4.6 | 594.1 | 4.7 |
| **M.m.p.** | – | – | 307.1 | 11.1 | 1592.6 | 12.5 |
| **M.m.s.** | 60.4 | 18.3 | 806.1 | 29.2 | 3951.6 | 31.0 |
| **ps-M.p.e.** | 9.4 | 2.8 | 135.3 | 4.9 | 1013.6 | 7.9 |
| **pi-M.p.e.** | 14.0 | 4.2 | 59.0 | 2.1 | 489.5 | 3.8 |
| **pa-M.p.i.** | 46.5 | 14.0 | 227.9 | 8.3 | 851.8 | 6.7 |
| **pp-M.p.i.** | | | 245.1 | 8.9 | 842.8 | 6.6 |
| **Total** | 332.3 | 100 | 2757.8 | 100 | 12762.2 | 100 |

Notes.

M.t.s., M. temporalis superficialis; pz-M.t.s., *M. temporalis superficialis pars zygomatica*; pm-M.t.p., *M. temporalis profundus pars medialis*; pl-M.t.p., *M. temporalis profundus pars lateralis*; M.m.p., *M. masseter profundus*; M.m.s., *M. masseter superficialis*; ps-M.p.e., *M. pterygoideus externus pars superior*; pi-M.p.e., *M. pterygoideus externus pars inferior*; pa-M.p.i., *M. pterygoideus internus pars anterior*; pp-M.p.i., *M. pterygoideus internus pars posterior*.

obtained from 3D-reconstructions, while in *M. tridactyla* they were larger, possibly due to soft tissue shrinking caused by the long period of staining of the latter (eight weeks; *Hedrick et al., 2018*). Below, we provide an anatomical description of the musculature of each of the three anteater species. Anatomical structures of the skull and mandible that are relevant to the description are depicted in Fig. 1. The origins and insertions of the masticatory muscles are figured for one species (i.e., *Tamandua tetradactyla*) (Fig. 2) to serve as a reference and to complement the 3D reconstructions. Three-dimensional surface models of the illustrated specimens (Fig. S1) are freely available at MorphoMuseumM (http://www.morphomuseum.com; *Ferreira-Cardoso et al., 2020*).

**Table 3** Facial-masticatory muscle volumes (mm³; left column) and percentages (right column) obtained from the 3D models of the segmentation of the contrast-enhanced specimens.

| Muscles | Volume in mm³/facial-mast. volume (%) | | | | | |
|---|---|---|---|---|---|---|
| | *C. didactylus* | | *T. tetradactyla* | | *M. tridactyla* | |
| pe-M.b. | 17.9 | 25.7 | 371.9 | 27.4 | 990.4 | 15.1 |
| pi-M.b. | 49.9 | 71.6 | 911.1 | 67.2 | 5589.4 | 85.0 |
| M.ma. | 1.94 | 2.8 | 73.7 | 5.4 | NA | – |
| Total | 69.7 | 100 | 1356.7 | 100 | 6579.8 | 100 |

Notes.

pe-M.b., *M. buccinatorius pars externa*; pi-M.b., *M. buccinatorius pars interna*; M.ma., *M. mandibuloauricularis*.

## Anatomical description
### *Cyclopes didactylus*
*Masticatory apparatus*

*M. masseter superficialis.* The *M. masseter superficialis* (M.m.s.; Figs. 3A, Figs. 3C and Figs. 3D) is the only muscle of the masseter muscle complex present in *C. didactylus*. The M.m.s. is anteroposteriorly elongated and originates from the lateral surface of the zygomatic process of the maxilla (Fig. 1B). The jugal bone is absent in *C. didactylus*. The origin of the M.m.s. consists of a long and strong posteroventrally projecting tendon that covers the most anterior half of the M.m.s. The muscle fibers of this anterior part are slightly obliquely oriented and compose the *pars anterior* of the M.m.s. (pa-M.m.s.; Fig. 3). The pa-M.m.s. inserts laterally from the posterior part of the dentary pad (Fig. 1C) to the anterior margin of the condyle (Fig. 3B). The pa-M.m.s. presents a *pars reflexa* inserting on the ventromedial margin of the ascending ramus of the mandible extending anteroposteriorly the level of the anterior margin of the coronoid process to the level of the mandibular canal. This part is covered laterally by the tendon from which it originates. Posteriorly, the M.m.s. presents a distinct *pars posterior* (pp-M.m.s.; Fig. 3; Fig. S2A) with anteroposteriorly oriented fibers. The pp-M.m.s. shares the origin with the pa-M.m.s. The former covers the pa-M.m.s. posteriorly to the coronoid process and inserts on the angular process of the mandible (Fig. 3). Its *pars reflexa* is continuous with the *pars reflexa* of the pa-M.m.s. and almost reaches the most posterior point of the angular process.

*M. masseter profundus* The *M. masseter profundus* is absent in *C. didactylus*.

*M. temporalis superficialis.* The *M. temporalis superficialis* (M.t.s.; Figs. 3A, 3B and 3C) is the largest of the four muscles of the temporal complex (Table 2). It is a fan-shaped muscle that originates from a scar along the dorsal edge of the temporal fossa (Fig. 1A, area in green). The temporal crest runs from the posterior end of the orbital ridge to the anterior surface of the root of the zygomatic process of the squamosal. A thick tendinous layer stretches from the origin of the M.t.s. and covers the posterodorsal part of the muscle. The M.t.s. is thinner at its origin and thicker at its insertion. The insertion is muscular on the dorsal tip and the dorsal part of the posterior margin of the coronoid process. An aponeurosis runs dorsoventrally along the anterior surface of the M.t.s. and completely covers the lateral and anterior surfaces of the coronoid process. The fiber fascicles of the

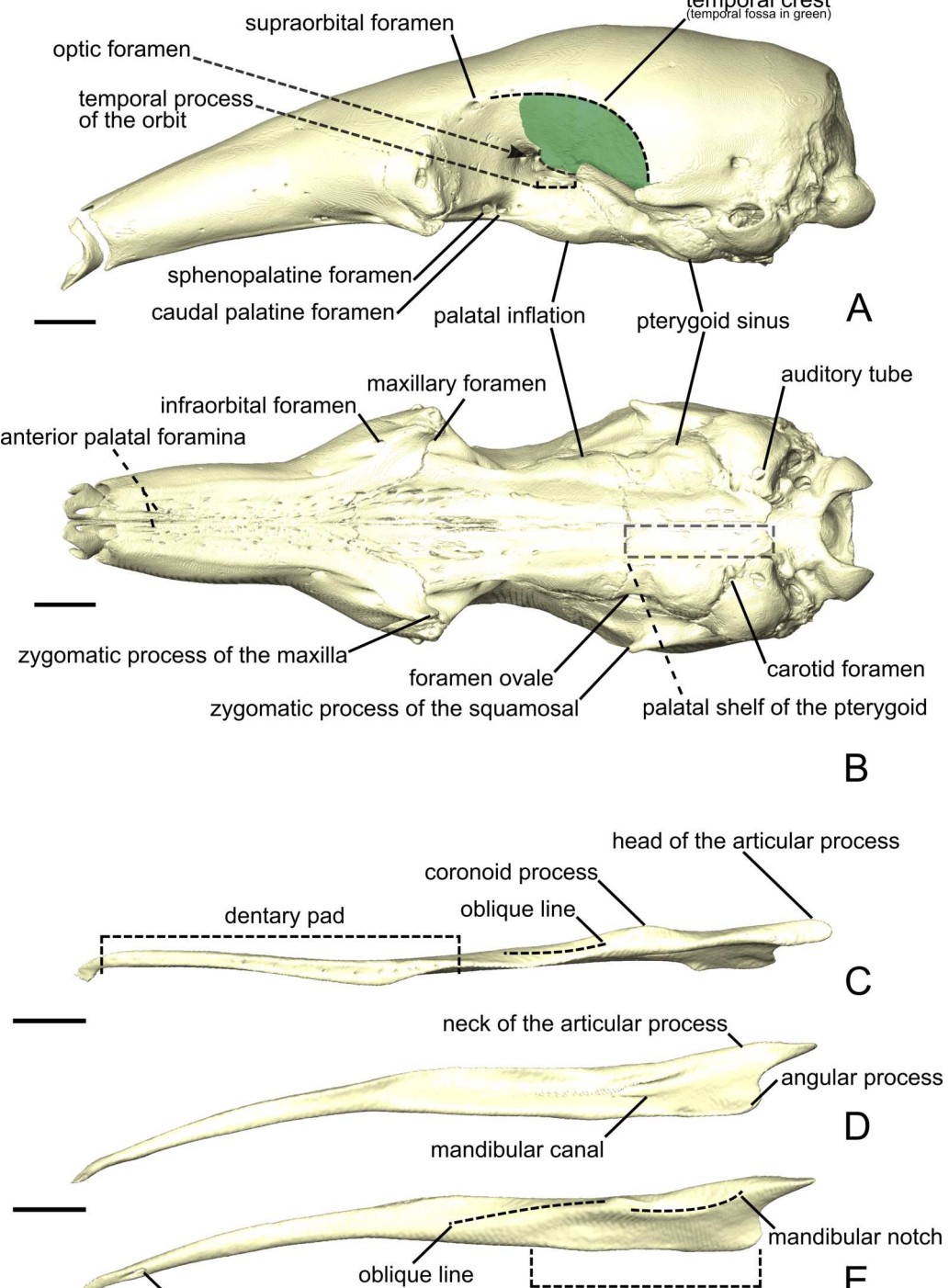

**Figure 1** **The skull (A, B) and mandible (C-E) of *Tamandua tetradactyla* shown in lateral (A) and ventral (B) views.** The area in green delimits the temporal fossa. The mandible is shown in dorsal (C), medial (D), and lateral (E) views. Anterior is to the left. Scale bar 10 mm.

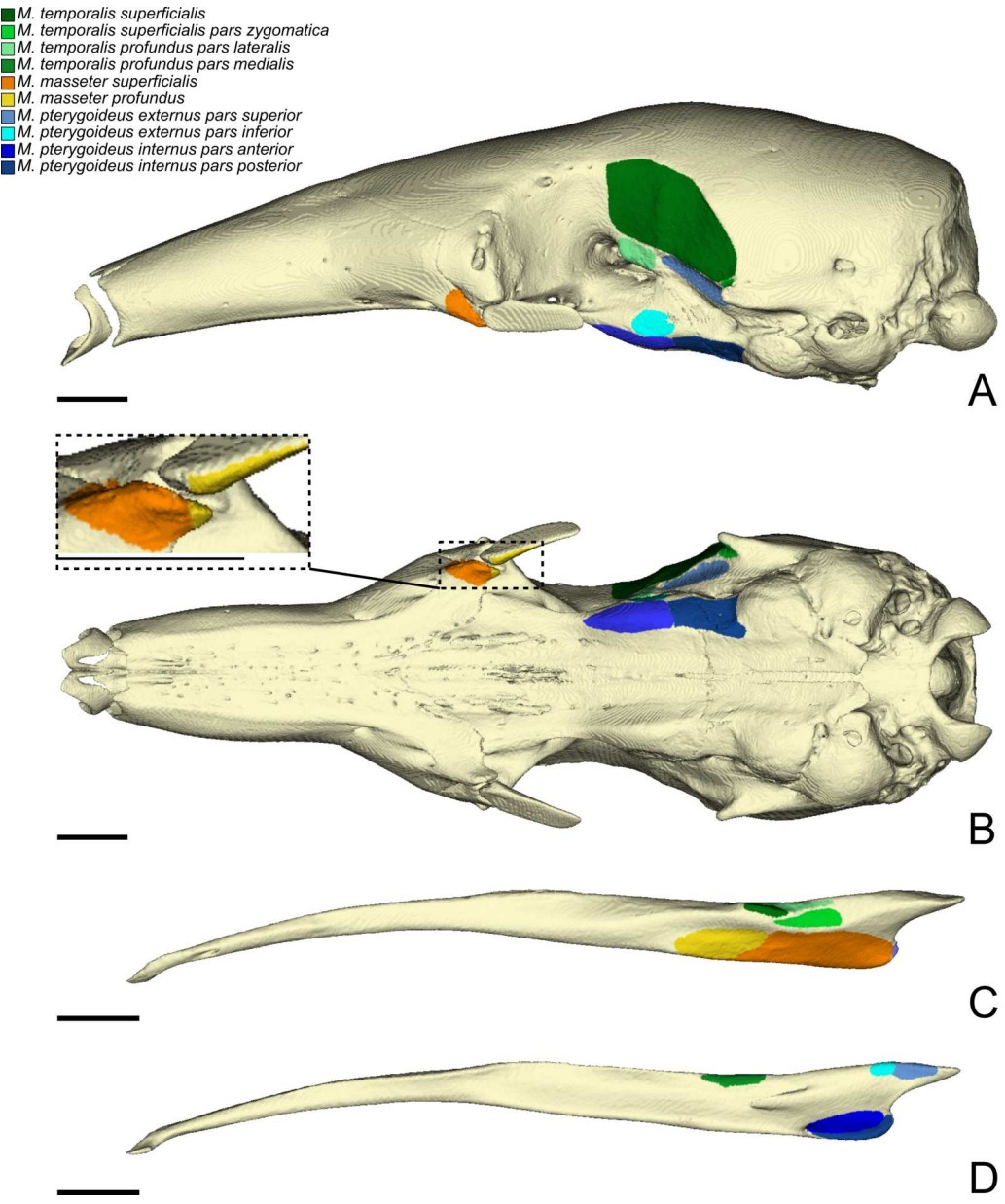

- **M. temporalis superficialis**
- **M. temporalis superficialis pars zygomatica**
- **M. temporalis profundus pars lateralis**
- **M. temporalis profundus pars medialis**
- **M. masseter superficialis**
- **M. masseter profundus**
- **M. pterygoideus externus pars superior**
- **M. pterygoideus externus pars inferior**
- **M. pterygoideus internus pars anterior**
- **M. pterygoideus internus pars posterior**

**Figure 2** **The skull (A, B) and mandible (C, D) of *T. tetradactyla* shown in lateral (A, C), ventral (B), and medial (D) views.** The colored areas represent the origin (A, B) and insertions (C, D) of the mastica­tory muscles. A color–coded legend is provided.

M.t.s. are organized in a bipennate structure (Fig. S2B). Deep fibers are dorsomedially oriented while superficial ones are dorsolaterally oriented. In cross-section, the insertion angle of medial fibers with the axis of pennation is about 26°, while lateral fibers present an angle of around 12°.

*M. temporalis superficialis pars zygomatica.* The *M. temporalis superficialis pars zygomatica* (pz-M.t.s.; Figs. 3A and 3C) is a relatively small muscle, which is well separated from the

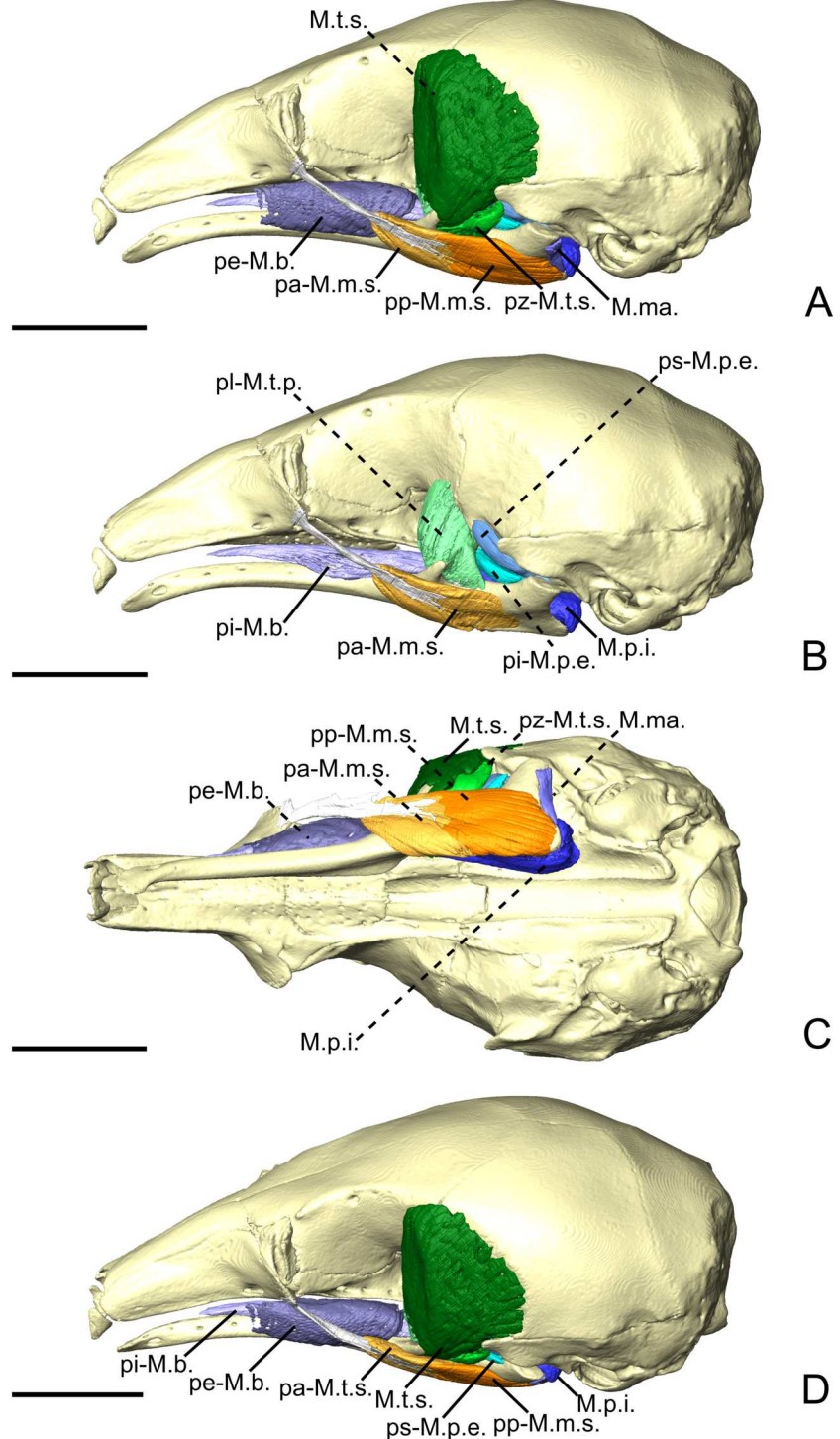

**Figure 3** **The masticatory and facial-masticatory musculature of *C. didactylus* in lateral (A, B), ventral (C), and dorsolateral (D) views.** Scale bar 10 mm. The more superficial muscles were removed in B. Muscle abbreviations as in Table 1.

M.t.s.. It originates from the ventromedial part of the zygomatic process of the squamosal and broadens ventrally to end on an anteroposteriorly elongated muscular insertion. The insertion occupies the lateral part of the mandibular notch. The pz-M.t.s. is wider dorsally and thinner ventrally, with fibers presenting an oblique orientation.

*M. temporalis profundus pars lateralis.* The temporal complex includes a deep component divided in two parts, the *M. temporalis profundus pars lateralis* pl-M.t.p.; (Figs. 3B, 4A and 4B) being the largest. The pl-M.t.p. takes its origin on a pseudo-elliptical area that extends from the posteroventral part of the orbital contribution of the frontal to the anteroventral part of the temporal fossa. The insertion of the pl-M.t.p. covers most of the posterolateral surface of the coronoid process, and narrows posteriorly along the mandibular notch. Contrary to the M.t.s., the pl-M.t.p. does not present a pennate structure, with fibers roughly vertically oriented.

*M. temporalis profundus pars medialis.* The *M. temporalis profundus pars medialis* (pm-M.t.p.; Figs. 4A and Figs. 4B) consists of the inner part of the M.t.p. that takes its origin from the orbit, between the ventral edge of the temporal fossa and the optic foramen. The pm-M.t.p. and the pl-M.t.p. are clearly separated posteriorly on the insertion, with the posterior tip of the pm-M.t.p. occupying a more ventromedial position at the level of the mandibular foramen. Fiber orientation and shape of the pm-M.t.p. is similar to that of pl-M.t.p., but the former's volume is about two thirds that of the latter. However, both muscles are anastomosed anteriorly.

*M. pterygoideus externus pars superior.* The *M. pterygoideus externus pars superior* (ps-M.p.e.; Figs. 3B, 4A and 4B) is a small anteroposteriorly elongated muscle. The ps-M.p.e. arises from a fossa that extends from the ventral part of the parietal, at the lower limit of the temporal fossa, into the glenoid fossa. It is the only part of the pterygoid muscle complex that takes its origin outside the pterygoid fossa. The muscle is mediolaterally compressed and obliquely oriented. Its posterior part presents a small torsion anterior to its ventrolateral projection towards the mandible. The insertion of the ps-M.p.e. consists of a small concavity just medioventral to the head of the articular condyle.

*M. pterygoideus externus pars inferior.* The *pars inferior* of the M.p.e. (pi-M.p.e.; Figs. 3B, 4A and 4B) consists of a short and fleshy muscle strap. The pi-M.p.e. originates from a small area on the sphenoid, laterally to the foramen rotundum, and dorsally adjacent to the origin of the *M. pterygoideus internus*. The muscle is mediolaterally wide and presents a more horizontal orientation than the ps-M.p.e. The pi-BE projects posterolaterally to insert on the anterior margin of the articular condyle, at mid-height. The medial part of the pi-M.p.e. projects posteriorly, inserting below the insertion area of the ps-M.p.e., reaching the mid-length of the head of the condyle.

*M. pterygoideus internus.* The *M. pterygoideus internus* (M.p.i.; Figs. 3, 4A, 4B and 5) arises from the pterygoid fossa and consists of a fleshy block that originates from the posterolateral part of the palatine to the level of the anterior margin of the ectotympanic

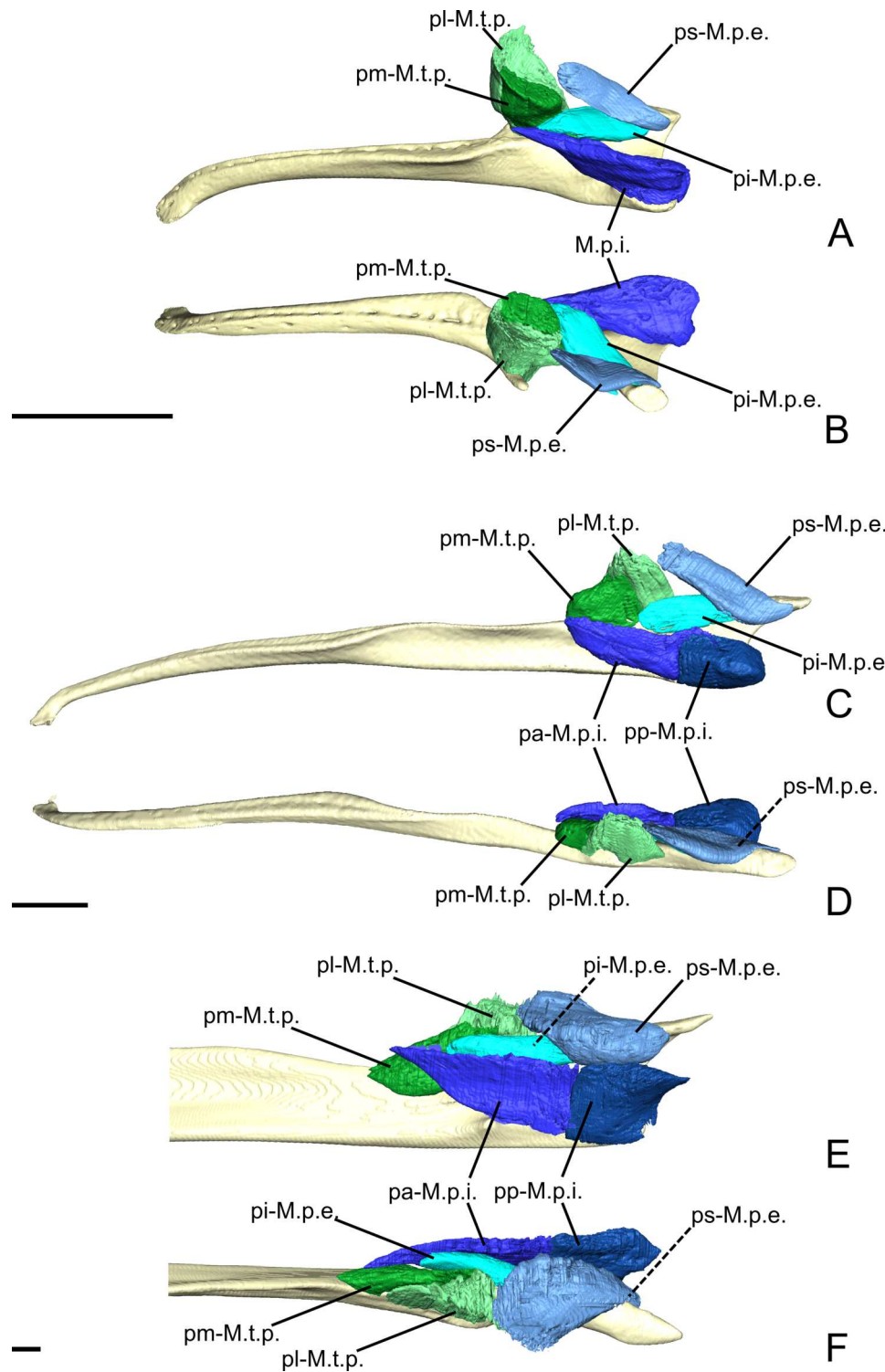

**Figure 4** **The *M. pterygoideus* and *M. temporalis profundus* muscle complexes of *C. didactylus* (A, B), *T. tetradactyla* (C, D), and *M. tridactyla* (E, F) in lateral (A, C, E) and dorsal (B, D, F).** E and F are zoomed on the ascending ramus. Scale bar 10 mm. Muscle abbreviations as in Table 1.

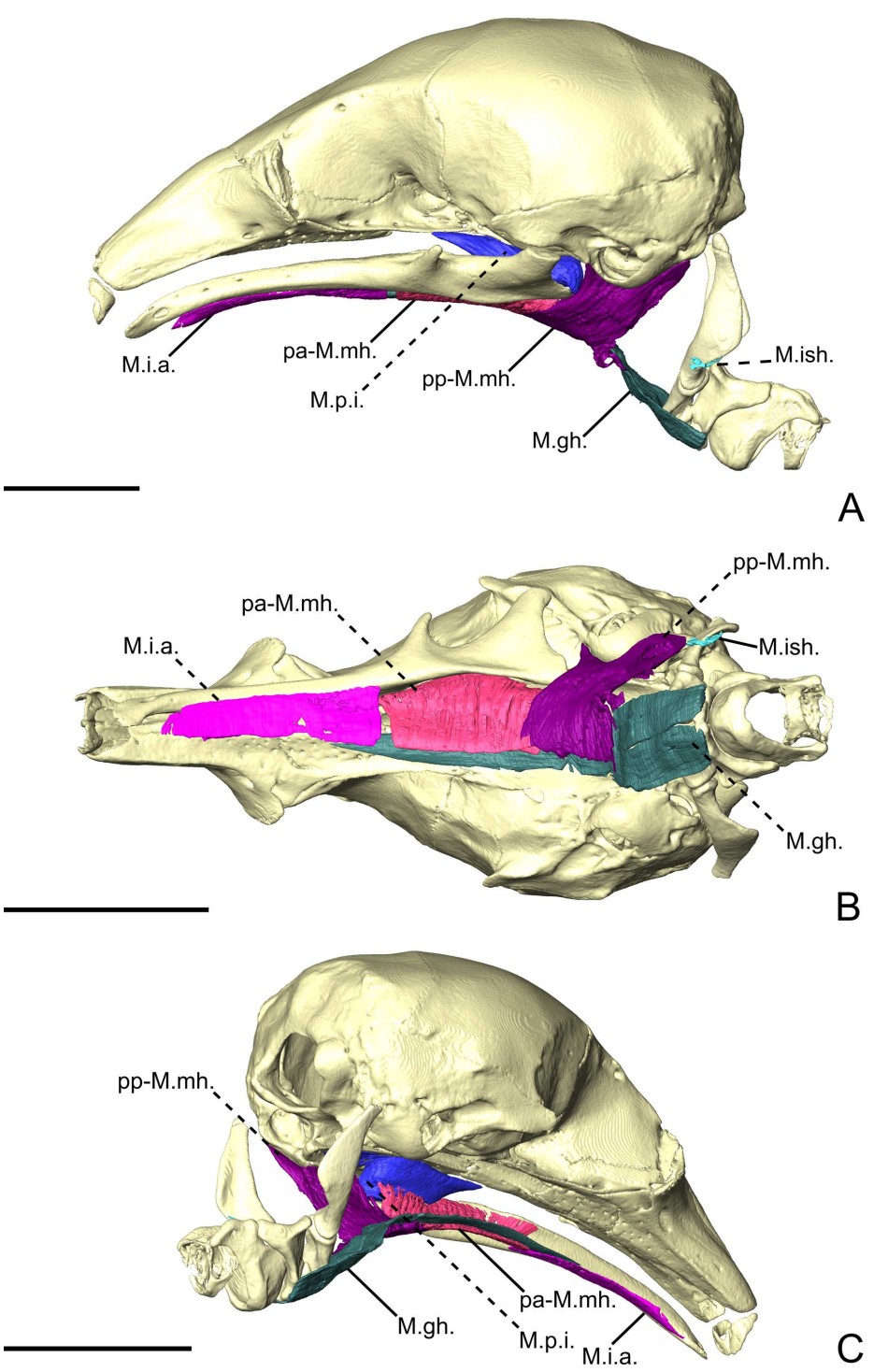

**Figure 5** **The intermandibular musculature, *M. geniohyoideus*, and *M. pterygoideus internus* of *C. didactylus* in lateral (A), ventral (B), and posteromedial (C) view.** Only the left half of the *M. intermandibularis* anterior is illustrated. A small vestige of the *M. interstylohyoideus* is also depicted. Scale bar 10 mm. Muscle abbreviations as in Table 1.

(Fig. 3A and 3B). Its fibers run anteroposteriorly with an oblique orientation and insert medially on the angular process of the mandible, from the level of the anterior margin of the head of the articular condyle to its posterior margin. In the most posterior part of their insertion, the fibers have a more posteroventral direction and form a small *pars reflexa* that wraps the posteriormost tip of the angular process. A dense connective tissue lies dorsal to the insertion of the M.p.i., posterior to the opening of the mandibular canal.

*Facial-masticatory musculature*

*M. buccinatorius pars externa.*   The *M. buccinatorius pars externa* (pe-M.b.; Figs. 3A, 3C and 3D) is distinguishable from the internal part of this muscle. It is a sheet-like muscle that envelopes the external surface of the *M. buccinatorius pars interna*, as well as the buccal salivary glands. Its origin stretches along the ventral edge of the maxilla and the palatine, from anteriorly to the inferior orbital foramen until the anterior part of the insertion of the *M. pterygoideus internus*. The ventral part of the pe-M.b. wraps the ventral portion of the *M. buccinatorius pars interna* (and the salivary glands, anteriorly) and attaches on a broad insertion area on the lateral surface of the mandible. The fibers have a dorsoventral orientation.

*M. buccinatorius pars interna.*   The *pars interna* of the *M. buccinatorius* muscle (pi-M.b.; Fig. 3B and Fig. 3D) is more voluminous when compared to the *pars interna*. The pi-M.b. originates from a thin fiber bundle posterior to the buccal commissure and is covered by the pe-M.b. just posteriorly. The pi-M.b. is bordered by the salivary glands, ventrally and laterally, anterior to the level of the sphenopalatine foramen. The pi-M.b. is a long muscle that reaches as far posteriorly as the level of the coronoid process. It is characterized by a buccal projection that sits between the upper and lower jaws (Fig. 3D). The lateral part of the pi-M.b. contacts the pe-M.b. and does not attach to any bone surface. Posteriorly, the pi-M.b. inserts on the dorsomedial surface of the mandible, along the fossa located between the posterior part of the dentary pad and the coronoid process. Its insertion ends posterior to the coronoid process where it contacts the *M. temporalis profundus pars medialis* and the anterior part of the *M. pterygoideus internus*. The fibers of the pi-M.b. are anteroposteriorly oriented.

*M. mandibuloauricularis.*   The *M. mandibuloauricularis* (M.ma.; Figs. 3A and 3C) is a strap-like bundle that takes its origin on the anteroventral part of the auricular cartilage. The M.ma. projects ventromedially to insert on the posterodorsal edge of the angular process of the mandible. The insertion is small and is located between the posterior parts of the masseteric and pterygoid fossae of the mandible. The M.ma. fibers presents a mediolateral orientation with a strong ventral component.

*Intermandibular musculature*

*M. intermandibularis anterior.*   The *M. intermandibularis anterior* (M.i.a.; Fig. 5) is a thin, dorsolaterally wide, and elongated muscle. *Naples (1999)* described this muscle as the anterior part of the *M. mylohyoideus pars anterior*. The M.i.a. takes its origin on the

cartilage of the unfused mandibular symphysis. The muscle has two insertions on the ventrolateral margin of both hemimandibles, wrapping around their ventral edges. In ventral view (Fig. 5B), it covers the anterior part of the base of the tongue and the anterior part of the *geniohyoideus* (Fig. 5). The M.i.a. extends posteriorly for about half the length of the mandible, its posterior end being clearly separated from the anterior margin of the *M. mylohyoideus pars anterior* (see below). Its fibers are transversely oriented and are continuous between mandibles, with this muscle consisting of one single element.

*M. mylohyoideus pars anterior.* The *M. mylohyoideus pars anterior* (pa-M.mh.; Fig. 5) consists of a fibrous sheet that originates ventrally to the dentary pad, on the medial surface of the mandible. This muscle is homologous to the *pars medius* of the *M. mylohyoideus* described by *Naples (1999)*. The origin area stretches from the widest point of the dentary pad to its posteriormost point. Posteriorly, its origin shifts from the mandible to the ventromedial surface of the *M. pterygoideus internus* (M.p.i.). At the posterior end of the M.p.i. the origin changes again, creating a dorsolateral gap separating the anterior and the posterior fibers. We consider this to be the posterior limit of the pa-M.mh., with the posterior part being considered the *M. mylohyoideus pars posterior*. The fibers are transversely oriented ventrally and insert along a fibrous midline raphe that connects the left and right pa-M.mh.s (as in Fig. S2C).

*M. mylohyoideus pars posterior.* The *M. mylohyoideus pars posterior* (pp-M.mh.; Fig. 5) is continuous with the pa-M.mh. The division between the two parts is set by the difference of the origin. The pp-M.mh. takes its origin on the ventromedial surface of the tympanic bulla, parallel to the auditory tube. The fibers display the same orientation as in the *pars anterior* and insert on a fibrous midline raphe. However, near the posterior end of the hard palate, the left and right muscles appear to anastomose in the midline, with the intertonguing contact becoming less spaced. As the *M. interstylohyoideus* (Fig. 5) and the posterior part of the *M. mylohyoideus pars posterior* were not preserved in our specimens of *C. didactylus*, the attachment of the pp-M.mh. to the hyoid system is not visible.

### *Tamandua tetradactyla*
#### Masticatory apparatus

*M. masseter superficialis.* The M.m.s. (Fig. 6A, Fig. 6C and Fig. 6D) is a fleshy, anteroposteriorly long muscle; its anterior and posterior ends are angular in shape in lateral view. The fibers of the M.m.s. are slightly oblique and take their origin on the lateral surface of the zygomatic process of the maxilla through a strong tendon. The M.m.s. inserts on the shallow masseteric fossa of the mandible. It covers most of the lateral surface of the ascending ramus, including most of the more anterior *M. masseter profundus* (see below). The M.m.s. is thicker posteriorly, and thins down anteriorly as it overlies the *M. masseter profundus*. The tendon of the M.m.s. was not visible in the contrast-enhanced specimen. The M.m.s. presents a *pars reflexa* that runs from the level of the posterior part of the jugal to the posterior tip of the angular process of the mandible. Anteriorly, the M.m.s. presents a small projection towards the zygomatic process of the mandible.

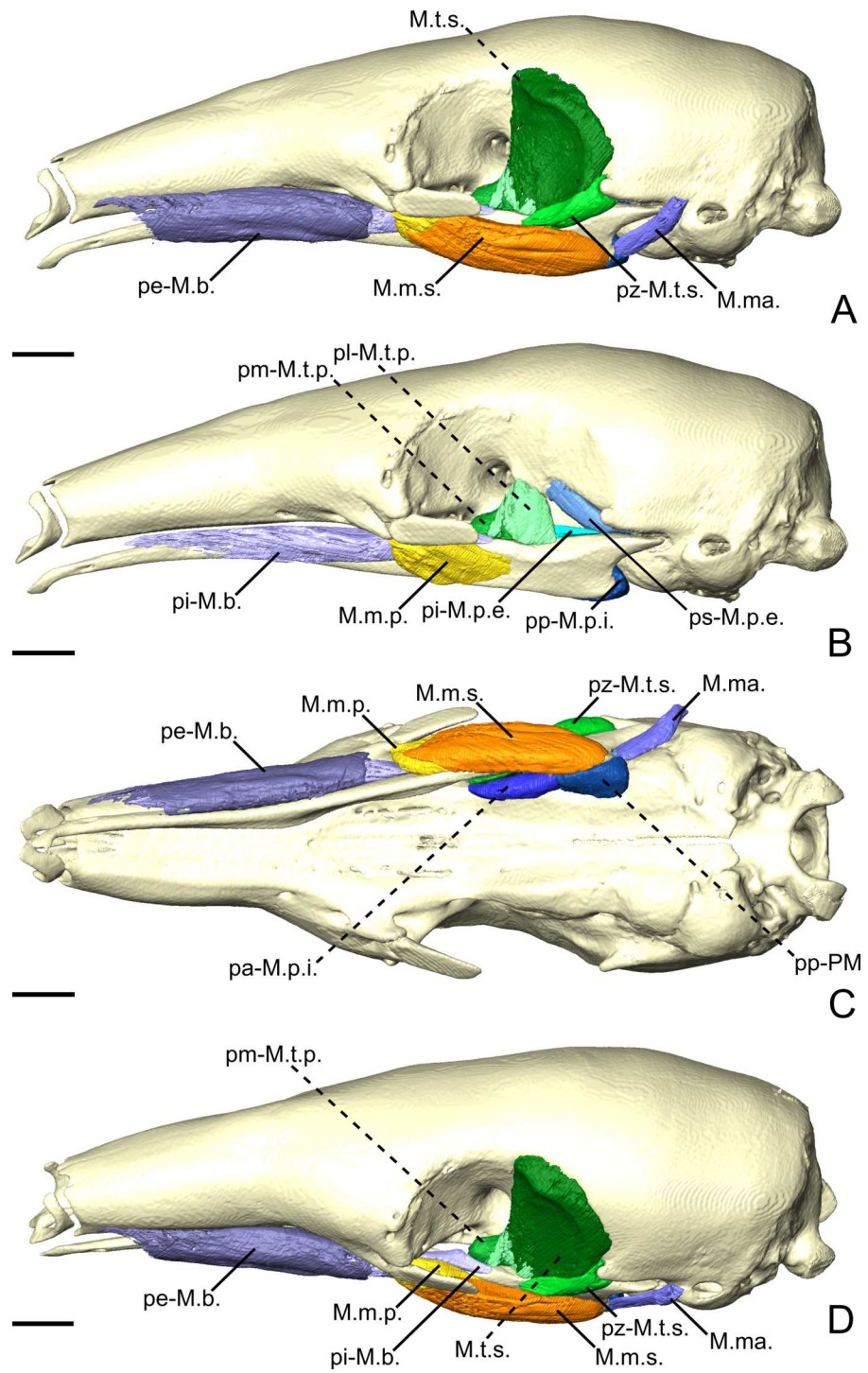

**Figure 6  The masticatory and facial-masticatory musculature of _T. tetradactyla_ in lateral (A, B), ventral (C), and dorsolateral (D) views. Scale bar 10 mm.** The more superficial muscles were removed in B. Muscle abbreviations as in Table 1.

*M. masseter profundus.*   The *M. masseter profundus* (M.m.p.; Fig. 6B–D) is smaller than its superficial counterpart (M.m.s.). It takes its origin on the anterior part of the ventromedial surface of the jugal bone. Anteriorly, its origin area includes the most posteroventral surface of the zygomatic process of the maxilla. The fibers of the M.m.p. run obliquely to insert posteroventrally on the lateral surface of the mandible. The fibers are more vertical than those of the M.m.s. The muscle presents components with slight lateral and anterior orientations. The insertion area on the mandible stretches from the coronoid process to the level of the oblique line. Contrary to the M.m.s., the M.m.p. is thicker at its origin than at its insertion.

*M. temporalis superficialis.*   The M.t.s. (Fig. 6A and Fig. 6D) is one of the three muscles that forms the temporal complex. It is also the largest, arising from a relatively large surface between the dorsal edge of the temporal fossa and the origin of the ps-M.p.e. (Fig. 6). It is wide and broad in lateral view, and transversely compressed. It presents a fan-like shape, the fibers converging ventrally towards the small and flat coronoid process. The M.t.s. is medial to a large lacrimal gland, which fills most of the temporal fossa. The lateral surface of the M.t.s. is covered by a thin tendinous layer. Ventrally, the M.t.s. inserts on the dorsomedial surface of the coronoid process via a large aponeurosis. The M.t.s. muscle fibers are oriented vertically in the anterior part of the muscle, and are more oblique posteriorly.

*M. temporalis superficialis pars zygomatica.*   The *pars zygomatica* of the M.t.s. (pz-M.t.s.; Figs. 6A, Figs. 6C and Figs. 6D) is a small fleshy strip on the ventral margin of the M.t.s. Unlike the M.t.s., the pz-M.t.s. originates on a small area limited to the ventral surface of the zygomatic process of the squamosal (Fig. 6). Its obliquely oriented fibers insert on the dorsolateral surface of the mandibular notch. While the insertion area and orientation of the fibers are distinct from the anterior part of the M.t.s., both muscles are anastomosed posteriorly to their mid-length.

*M. temporalis profundus pars lateralis.*   The M.t.p. (Figs. 4C, 4D and 6B) is divided into two distinct parts. The *pars lateralis* (pl-M.t.p.; Figs. 4C, Figs. 4D and 6B) is a small fleshy block deep to the larger M.t.s. The pl-M.t.p. takes its origin from the crest formed between the anteroventral border of the temporal fossa and the groove for the ophthalmic vein and the oculomotor nerve (III) (orbital process). The pl-M.t.p. transversely widens from its origin to its insertion. Fiber orientation is similar to that of the anterior part of the M.t.s., although slightly more oblique in coronal view. The insertion of the pl-M.t.p. is short and extends from the mid-length of the mandibular notch to the anterior part of the coronoid process. It covers most of the dorsal surface of the mandible in width. While the insertion is mostly muscular, the pl-M.t.p. shares the aponeurosis with the M.t.s. anteriorly.

*M. temporalis profundus pars medialis.*   The pm-M.t.p. (Figs. 4C, Figs. 4D, 6B and 6D) is the smallest part of the temporal muscle complex. It has no insertion, as it anastomoses with the pl-M.t.p. posterolaterally, but both parts could be easily separated during dissection. The fibers of the pm-M.t.p. are vertically oriented. Their insertion is medial to that of the

pl-M.t.p. and extends from the level of the anterior tip of the pi-M.p.e. to the anterior margin of the optic foramen. The medialmost part of the pm-M.t.p. wraps the mandible medially to insert on its dorsomedial surface; it contacts the dorsal part of the pa-M.mh. (see 'Intermandibular musculature').

*M. pterygoideus externus pars superior.*   The ps-M.p.e. (Figs. 4B, 4C, 4D and 6B) is a strap-like muscle that arises from an elongated fossa along the ventral limit of the temporal fossa. Its obliquely oriented fibers run posteriorly to medially wrap around the head of the articular condyle of the mandible (Figs. 4C and 4D). The insertion extends from the anterior part to the posterior tip of the blunt articular condyle. The ps-M.p.e. overlies the insertion of the pi-M.p.e. (see below).

*M. pterygoideus externus pars inferior.*   Similarly to the ps-M.p.e., the pi-M.p.e. (Figs. 4C and 4D) has a strap-like shape. In contrast with its upper counterpart, the pi-M.p.e. takes its origin on the pterygoid fossa. Specifically, the origin of the pi-M.p.e. is a small flattened area on the lateral surface of the palatal inflation. Its fibers are obliquely oriented and insert dorsally on the neck of the condylar process of the mandible.

*M. pterygoideus internus pars anterior.*   The M.p.i. is divided into two distinct parts. The *pars anterior* (pa-M.p.i.; Figs. 4C, 4D and 6C) takes its origin on the lateral and ventrolateral surfaces of the palatine sinus. The origin is muscular and spans from level of the caudal palatine foramen to an area just posterior to the origin of the pi-M.p.e., near the posterior limit of the palatal inflation. The fibers are more oblique anteriorly than posteriorly, and insert on the dorsal part of the pterygoid fossa of the mandibular ascending ramus. The posterior part of the pa-M.p.i. is thinner than the anterior part. The thick portion of the pa-M.p.i. serves as an attachment area for a small anterior projection of the pp-M.mh. (see 'Intermandibular musculature').

*M. pterygoideus internus pars posterior.*   The pp-M.p.i. (Figs. 4C, 4D, 6B and 6C) consists of a fleshy block that takes its origin on an area located between the posterior part of the palatal inflation and the small fossa anterior to the pterygoid sinus. A coronal section shows that the fibers are obliquely oriented (Fig. S2D). The pp-M.p.i. presents a very small *pars reflexa* that extends from the anterior- to the posteriormost part of the pterygoid fossa of the ascending ramus, wrapping around the margin of the small angular process (Figs. 4C and Figs. 4D).

*Facial-masticatory musculature*

*M. buccinatorius pars externa.*   The pe-M.b. (Fig. 6A, 6C and 6D) is a thin sheet of obliquely oriented muscle fibers that envelops the pi-M.b. and the buccal salivary glands. The muscle takes its narrow and anteroposteriorly elongated origin on the maxilla. Its posterior limit attaches just anteroventral to the zygomatic process of the maxilla. Its anterior part consists of a thin strap on the lateral surface of the maxilla, close to the lateral limit of the nasal cavity. The muscle wraps around the pi-M.b. and reflects medially to insert along the dorsal part of the lateral surface of the mandible. Its insertion is shorter than its origin, extending

from the level of the infraorbital foramen for the posterior two thirds of the length of the horizontal ramus.

*M. buccinatorius pars interna.*   The pi-M.b. (Fig. 6B) is an elongated and fleshy muscle that takes its origin just posterior to the buccal commissure on the ventral part of the lateral surface of the maxilla. The anterior part of the pi-M.b. has a thin projection of its dorsal part that wraps around the lateral border of the dentary pad, to project into the space between the upper and lower jaws. This part of the muscle contacts the salivary glands ventrolaterally. The pi-M.b. lateral surface is enveloped by the pe-M.b. anterior to the zygomatic process of the maxilla. The muscle fibers are horizontally oriented. Posteriorly, the pi-M.b. inserts on the dorsal surface of the mandible, at the level of the optic foramen. The insertion is laterally adjacent to that of the pm-M.t.p. It extends anteriorly to reach the level of the maxillary foramen. The orbital part of the pi-M.b. is flattened due to the presence of the large lacrimal gland, dorsally. Madially, it is limited by the presence of the pm-M.t.p.

*M. mandibuloauricularis.*   The M.ma. (Figs. 6A, 6C, and 6D) is a small fleshy muscle with a pseudocylindrical shape. It takes its origin on the anteroventral part of the auricular cartilage. The M.ma. narrows ventrally towards its insertion on a small area of the posterodorsal margin of the angular process of the mandible, between the insertions of the M.m.s. and the M.p.i. The M.ma. presents dorsoventrally directed fibers with a slight medial component.

*Intermandibular musculature*

*M. intermandibularis anterior.*   The M.i.a. (Fig. 7; pa-M.mh. *sensu Naples, 1999*) is a sheet-like muscle that arises from the symphysial cartilage. The M.i.a. fibers are transversely oriented. They insert on both hemimandibles, covering the base of the tongue and the tendon of the *geniohyoideus* in ventral view (Fig. 7). The M.i.a. is, therefore, a single muscle with no bilateral counterpart (Fig. S2E). It wraps around the ventral margin of the mandible to insert just dorsal to it, on the lateral surface. The M.i.a. extends posteriorly for slightly more than half the length of the horizontal ramus of the mandible. Posteriorly, it is adjacent to the anterior margin of the pa-M.mh.

*M. mylohyoideus pars anterior.*   The pa-M.mh. (Fig. 7) is a sheet-like muscle with transversely oriented fibers, and covers the base of the tongue and the long tendon of the *geniohyoideus* (M.gh., not described). Its morphological similarities with the M.i.a. caused previous studies to describe the latter as a distinct part of the *mylohyoideus* complex (*Naples, 1999*). In contrast to the M.i.a., the pa-M.mh. insertion takes its origin on the ventral part of the medial surface of the mandible, between the widest point of the dentary pad and the pterygoid fossa posteriorly (Fig. 7). In addition to a different insertion, the pa-M.mh. is a bilaterally symmetric element, with both counterparts united medially by a small layer of conjunctive tissue (Fig. S2C). The pa-M.mh. is slightly thicker than the M.i.a. Posteriorly, the pa-M.mh. anastomoses with the pp-M.mh., the two parts being

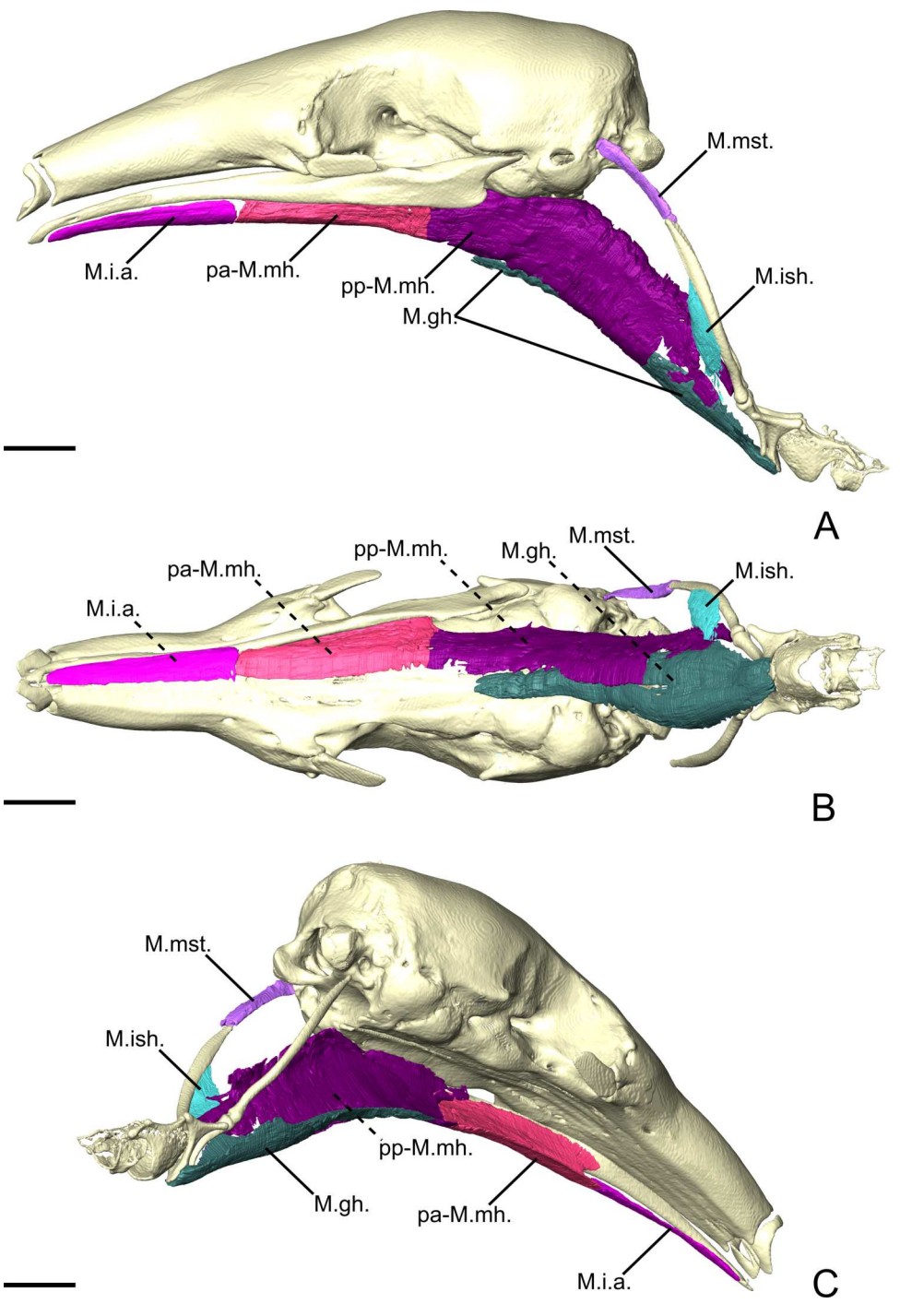

**Figure 7** **The intermandibular musculature, *M. geniohyoideus*, *M. interstylohyoideus*, and *M. mastostyloideus* of *T. tetradactyla* in lateral (A), ventral (B), and posteromedial (C) view.** Only the left half of the *M. intermandibularis* anterior is illustrated. Scale bar 10 mm. Muscle abbreviations as in Table 1.

continuous. In coronal view, the division between the two muscles is characterized by the passage of the sublingual artery (*Evans & De Lahunta, 2013*), ventral to the pa-M.p.i. (Fig. S2F).

*M. mylohyoideus pars posterior.*   The *pars posterior* of the *M. mylohyoideus* (pp-M.mh.; Fig. 7) is broader than pa-M.mh. At the level of the orbital fissure, the sublingual artery (*Evans & De Lahunta, 2013*) splits the insertions of the pa-M.mh. and the pp-M.mh. While the pa-M.mh. inserts on the mandibular ramus, the insertion of the pp-M.mh. extends along the medial surface of the palatine inflation, then along the ventromedial surface of the pterygoid sinus to continue posteriorly to the level of the auditory tube (Fig. 7). Additionally, a thin muscular projection inserts on the medial surface of the pa-M.p.i. Posterior to the hard palate, the pp-M.mh. inserts on the soft palate, keeping its shape until it reaches the anterior part of the *M. stylopharyngeus* (not described), where it bifurcates. A fleshy fiber extension projects posteriorly to attach on a small area of the anterior surface of the stylohyal, just dorsal to its suture with the epihyal. On the other hand, a ventral sheet-like projection attaches to the tendon of the *M. interstylohyoideus* (M.ish., not described; Fig. 7). As in other cases, the tendon could not be segmented. Nevertheless, the presence of muscular fibers of the M.ish. confirm the position of the insertion of the pp-M.mh. described in previous studies (*Reiss, 1997*).

### *Myrmecophaga tridactyla*
*Masticatory apparatus*

*M. masseter superficialis.*   In *M. tridactyla*, the M.m.s. (Figs. 8A,8C and 8D) is a fleshy and anteroposteriorly elongated muscle. The M.m.s. originates from the ventrolateral margin of the zygomatic process of the maxilla. A strong tendon connects the origin to the almost horizontally oriented muscular fibers. The M.m.s. is thin at the origin, as it overlies the posterior part of the M.m.p. It thickens posteriorly, as it extends anteriorly to the lacrimal foramen and the posterior part of the masseteric fossa. The M.m.s. presents a *pars reflexa* throughout most of its length (Fig. 8C). The *pars reflexa* wraps around the ventral edge of the mandible and becomes larger posteriorly, covering only the very posteroventral tip of the small angular process (Figs. 8A and 8C).

*M. masseter profundus.*   The M.m.p. (Figs. 8B, 8C and 8D) takes its origin on the anterior part of the ventrolateral surface of the zygomatic arch. Its area of origin includes the small jugal bone and the posteroventral surface of the zygomatic process of the maxilla. The M.m.p. is in contact with the posterior part of the pi-M.b., medially (Fig. 8D). The M.m.p. is obliquely oriented; it inserts ventrally on the mandible and presents a small *pars reflexa*. The muscle is thick at its origin but thins down posteriorly, where it is overlain by the M.m.s. The M.m.p. is half the length of the M.m.s., with its insertion area stretching from the most anterior part of the masseteric fossa to near the level of the coronoid process.

*M. temporalis superficialis.*   The M.t.s. (Figs. 8A and 8D) is a flat muscle covered almost entirely by the large lacrimal gland. It is a fan-like muscle originating from the temporal

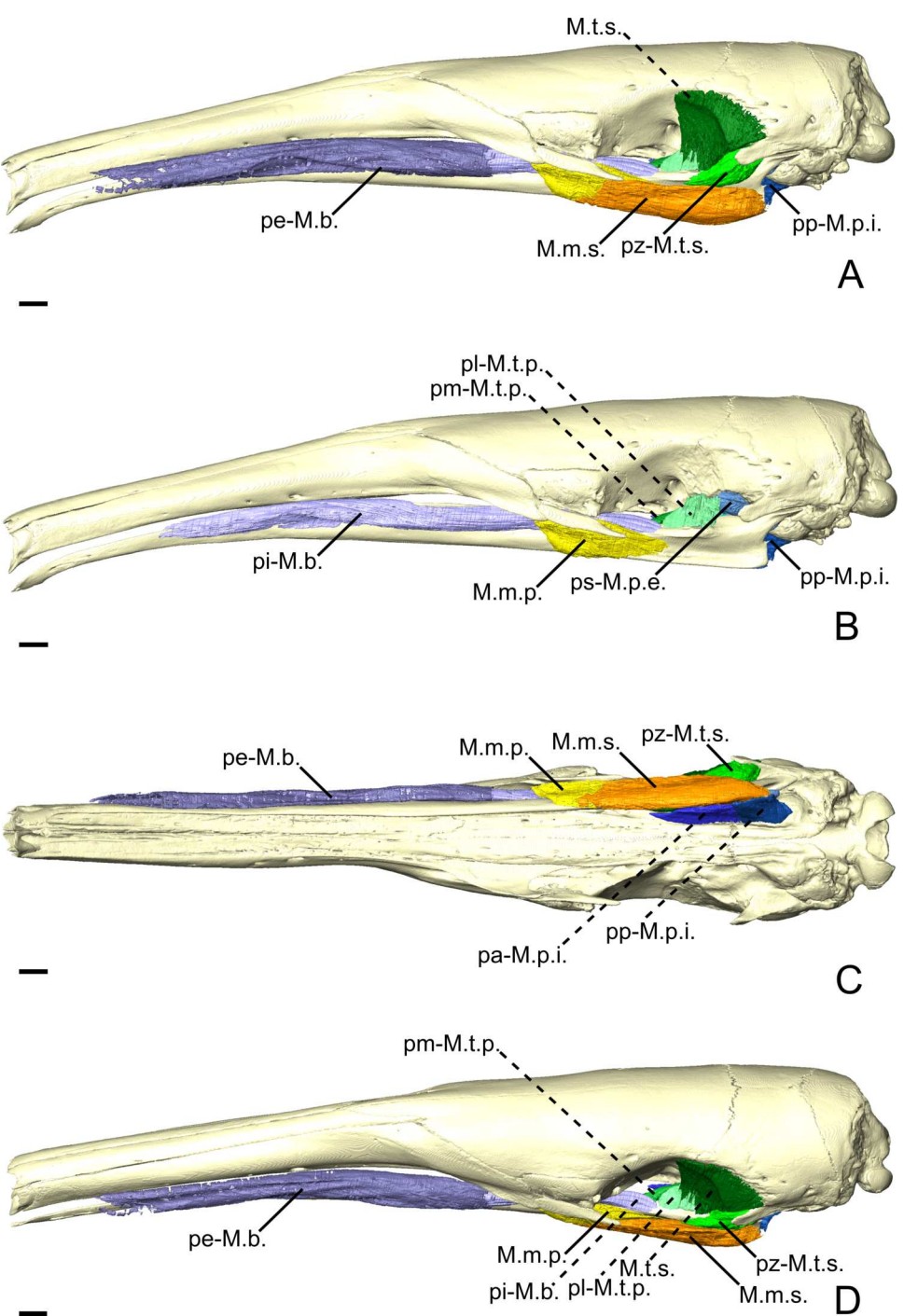

**Figure 8 The masticatory and facial-masticatory musculature of *M. tridactyla* in lateral (A, B), ventral (C), and dorsolateral (D) views.** Scale bar 10 mm. The more superficial muscles were removed in B. Muscle abbreviations as in Table 1.

fossa extending from the level of the optic foramen to the root of the zygomatic process of the squamosal. The lateral surface of the M.t.s. is covered by a tendinous layer that thickens ventrally near the insertion of the muscle on the small coronoid process. While the ventrally converging fibers of the M.t.s. reach the coronoid process posteriorly, the anterior part of the muscle inserts on the mandible uniquely *via* its tendinous layer (Fig. 8A). The M.t.s. is well-separated from the *pars zygomatica*, posteriorly, due to the very distinct orientation of the muscular fibers.

*M. temporalis superficialis pars zygomatica.* The *pars zygomatica* of the M.t.s. (pz-M.t.s.; Figs. 8A, 8C, and 8D) is a fleshy and thick part of the M.t.s. complex. It arises from the medial and posteroventral surfaces of the zygomatic process of the squamosal and extends anteroventrally with an oblique orientation. The pz-M.t.s. displays a medial portion that extends along the anterior margin of the neck of the mandibular articular process and inserts on the posterior surface of the blunt coronoid process. The lateral part of the pz-M.t.s. is larger and extends along the surface lateral to the mandibular notch. The most ventral part of the pz-M.t.s. is slightly overlain by the dorsal margin of the M.m.s. The pz-M.t.s. is easily distinguishable from its larger counterpart due to the different orientation angle of its fibers.

*M. temporalis profundus pars lateralis.* The M.t.p. (Figs. 4E, 4F, 8B and 8D) is divided into medial and lateral parts. The *pars lateralis* (pl-M.t.p.) is a blocky-shaped muscle arising from the ventral limit of the temporal fossa between the anterior tip of the orbital process and the insertion of the ps-M.p.e. The posterior part of the pl-M.t.p. presents a quadrangular shape in lateral view, with the anterior part tapering in near the pi-M.b. The muscular fibers are dorsoventrally oriented with an oblique transversal component. The pl-M.t.p. inserts on the dorsal surface of the ascending ramus deep to the insertion of the M.t.s. While the M.t.p. is well separated from the M.t.s. during the classical dissection, the incomplete staining of the former makes it sometimes hard to delimit. Anteriorly, the insertion of the pl-M.t.p. extends until the level of the anterior margin of the optic foramen (Fig. 8B).

*M. temporalis profundus pars medialis.* The pm-M.t.p. (Figs. 4E, 4F, 8B and 8D) in *M. tridactyla* is a medioventrally extending projection of the pl-M.t.p. Both parts are anastomosed posteriorly, sharing the medial part of the M.t.p. origin. The pm-M.t.p. arises from the ventral surface of the orbital process lateral to the orbital fissure and the foramen rotundum. Slightly anterior to its origin, the pm-M.t.p. extends ventrally on the lateral surface of the ascending ramus (Figs. 4E and 4F). Anterior to this point, the two parts of the M.t.p. are distinguished by different insertion areas (Figs. 4E and Figs. 4F), with pm-M.t.p. reflecting medially. The insertion of the pm-M.t.p. is broad and extends ventrally almost until the level of the mandibular canal. It is limited posteriorly by the mandibular canal. The pm-M.t.p. tapers anteriorly to its contact with the posterior part of the pi-M.b. at the orbit mid-length. Fiber orientation in the pm-M.t.p. is similar to that of the pl-M.t.p.

*M. pterygoideus externus pars superior.* The ps-M.p.e. (Figs. 4E, 4F and 8B) is a broad and wide fleshy sheet muscle arising from the large fossa extending from the anteroventral part of the squamosal to the ventral part of the temporal fossa. Its fibers are obliquely oriented and extend posteroventrally to insert on the mandible just anterior to the jaw joint. The posteroventral part of the ps-M.p.e. is characterized by a large *pars reflexa* that wraps around the medial edge of the articular process. The *pars reflexa* of the ps-M.p.e. overlays the posterior part of the *pars inferior* of the M.p.e.

*M. pterygoideus externus pars inferior.* The pi-M.p.e. (Figs. 4E and 4F) is a strap-shaped muscle that originates from the anterior part of the pterygoid fossa, at the level of the optic foramen. In contrast with the ps-M.p.e., the pi-M.p.e. is narrow and elongated. Its origin is thin and lies medial to the pm-M.t.p. The anterior part of the pi-M.p.e. is in tight contact with the pa-M.p.i. The pi-M.p.e. slightly thickens up posteriorly, assuming a circular cross-section. The muscular fibers are horizontally oriented, with an oblique component as they insert posterolaterally on the anterior part of the neck of the articular process (Figs. 4E and 4F). The insertion of the pi-M.p.e. reaches about half the length of the neck and is overlain laterally by the *pars reflexa* of the ps-M.p.e.. The pi-M.p.e. merges with the *pars reflexa* of the ps-M.p.e. by a thick band of connective tissue.

*M. pterygoideus internus pars anterior.* The *pars anterior* (pa-M.p.i.; Figs. 4E, 4F and 8C) is the larger of the two parts of the M.p.i. It takes its origin from the small crest formed by the lateral edge of the palatine. In lateral view, the pa-M.p.i. presents a pseudorectangular shape. Anteriorly, the muscle narrows down (Figs. 4E and 4F). The most anterior fibers originate just anterior to the level of the optic foramen. The fibers extend ventrally to insert on a lateral prominence of the mandibular ascending ramus, ventral to the passage of the inferior alveolar nerve and artery. Posteriorly, the fibers are dorsoventrally oriented, with an oblique transverse component. Both origin and insertion of the pa-M.p.i. end roughly at the level of the pterygopalatine suture.

*M. pterygoideus internus pars posterior.* The smallest component of the M.p.i. is a fleshy pseudorectangular band in lateral view (Figs. 4E and 4F). The origin of the pp-M.p.i. (Figs. 4E, 4F and 8C) is very thin and extends from near the palatine-pterygoid suture to the pterygoid sinus at the level of the posterior limit of the jaw joint. The pp-M.p.i. is the continuation of the pa-M.p.i. until the tip of the angular process, where it reaches the insertion area of the M.ma. In lateral view, the fibers are vertically oriented, with a transversal component of about 21° relative to the sagittal axis of the skull. Posteriorly, the pa-M.p.i. becomes thicker but it tapers off abruptly at the level of the pterygoid sinus.

*Facial-masticatory musculature*

*M. buccinatorius pars externa.* The pe-M.b. (Figs. 8A, 8C, and 8D) is an extremely thin sheet enveloping the much thicker *pars interna* (see below) and the buccal salivary glands. The fibers of the pe-M.b. have an oblique orientation, arising from the long and extremely narrow origin on the maxilla. The origin extends from the level of the most posterior

mental foramen to the anterior edge of the zygomatic process of the maxilla. The pe-M.b. extends ventrally, envelopes the pi-M.b. and reflects medially. The muscle wraps around the ventromedial margin of the *pars interna* of the *M. buccinatorius* and projects dorsally to insert on the dorsolateral surface of the mandibular horizontal ramus. Its insertion and origin areas are similar in length, but the bad preservation of the soft tissues in the snout did not permit to clearly observe the anterior tip of its origin.

*M. buccinatorius pars interna.* The pi-M.b. (Fig. 8B) is extremely long anteroposteriorly, reflecting the elongation of the rostrum. The muscle takes its origin on the maxilla, adjacent to the labial commissure of the mouth, although the muscular fibers arise more posteriorly. The pi-M.b. fibers go on to insert on the dorsal surface of the horizontal ramus of the mandible, ventral to the eye and the lacrimal gland. The fibers have an almost horizontal orientation, leaning slightly ventrally. In cross section, the anterior part of the pi-M.b. is dorsoventrally elongated. The most anterior part of the pi-M.b. presents a medial flap-like projection that rests between both jaws (Fig. 8B). This part of the pi-M.b. contacts the salivary glands laterally. At the length of the posterior most tip of the nasal, the pi-M.b. drifts ventrally and narrows dorsoventrally (Fig. 8B). Posterior to the zygomatic process of the maxilla, the pi-M.b. leans medially to a position between the jaws, deep to the M.m.s. This marks the beginning of the insertion of the pi-M.b., which extends to the anterior part of the insertion of the M.t.p., just anterior to the level of the optic foramen.

*M. mandibuloauricularis.* The M.ma. consists of a small fiber bundle that takes its origin from the anterior part of the auricular cartilage. It inserts on the posterior tip of the angular process, between both the pp-M.m.s. and pp-M.p.i. This muscle was damaged on the digitally dissected side of the skull and was described based on its right counterpart.

*Intermandibular musculature*

*M. intermandibularis anterior.* The M.i.a. (Fig. 9; pa-M.mh. *sensu Naples, 1999*) is extremely elongated, extending for almost half the mandibular length (127.4 mm). This muscle is very thin and forms a sheet covering the tendon of the *M. geniohyoideus* as well as the tongue (not figured). Each fiber is attached to thin areas on the ventrolateral surfaces of both mandibles. The muscle wraps around the ventral margin of the mandible and stretches transversely to insert on the opposite side's hemimandible. The fibers are continuous between mandibles.

*M. mylohyoideus pars anterior.* The postcranial muscles in our specimen were not successfully stained by the iodine solution and, therefore, could not be illustrated and described (Fig. 9). The pa-M.mh. (Fig. 9; pm-M.mh. sensu *Naples, 1999*) is only partially stained and thus not completely represented in our 3D reconstructions. This muscle forms a thick sheet ventral to the tongue musculature. Its fibers are transversely oriented, connecting a midline of connective tissue to the medial surface of the mandible (Fig. 9). Both symmetric counterparts of the pa-M.mh. unite in the midline, but could be easily distinguished both during the classical and digital dissections. The pa-M.mh. is clearly

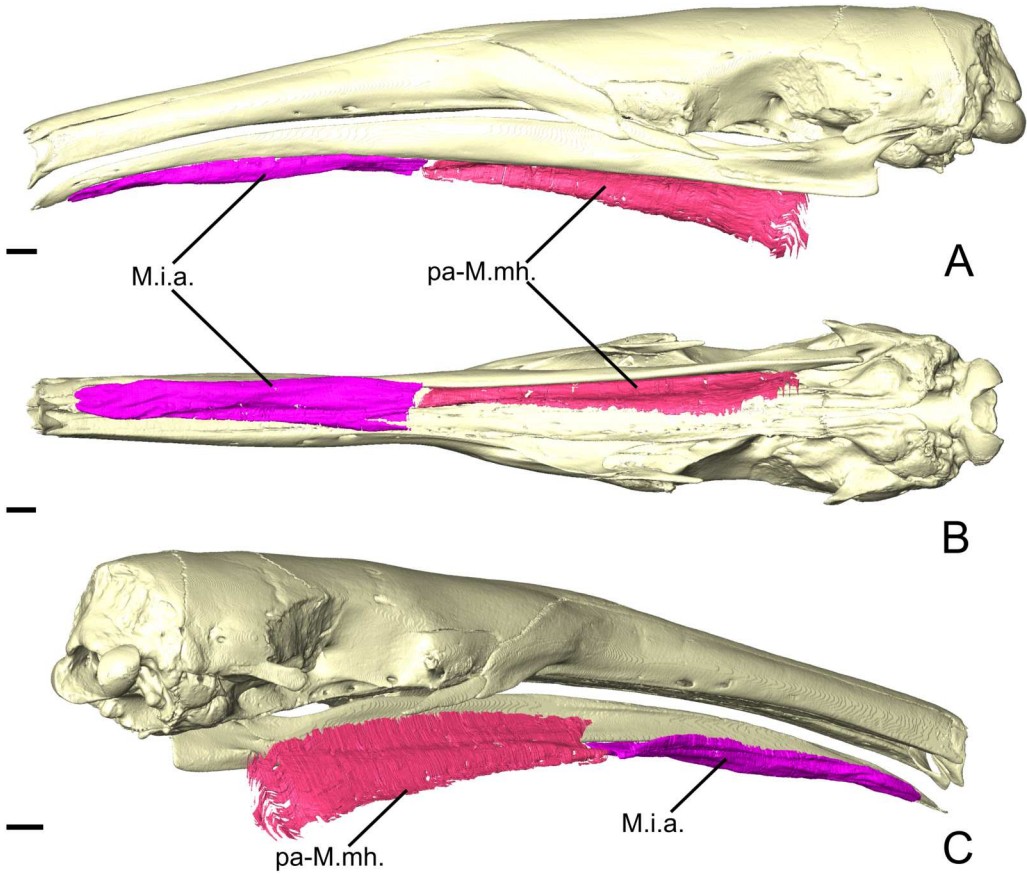

**Figure 9** **The intermandibular musculature of *M. tridactyla* in lateral (A), ventral (B), and posteromedial (C) view.** Scale bar 10 mm. Muscle abbreviations as in Table 1.

separated from the pp-M.mh. by a shift in the insertion from the mandible to the skull. The posterior end and the transition between the pa-M.mh. and the pp-M.mh. could not be segmented during the digital dissection.

## DISCUSSION

### Myological features and anteater systematics

External morphology has, for a long time, provided elements allowing extant anteaters to be split into two distinct groups (*Pocock, 1924*; *Reeve, 1940*; *Hirschfeld, 1976*; *Patterson et al., 1992*). Pygmy anteaters (*Cyclopes* spp.) are ascribed to a monogeneric family (Cyclopedidae) while tamanduas (*Tamandua* spp.) and the giant anteater (*Myrmecophaga tridactyla*) form the Myrmecophagidae (Fig. 10; *Gibb et al., 2016*). Although all anteaters present toothless and elongated jaws, this elongation is particularly pronounced in mymecophagids, reaching extreme proportions in the giant anteater (*M. tridactyla*). Pygmy anteaters present a shorter snout, a concave curvature of the basicranial/basifacial axis (*Gaudin & Branham, 1998*), pterygoids that do not meet in the midline, as well as relatively well-developed coronoid and angular processes of the mandible (*Hirschfeld, 1976*; *Engelmann, 1985*).

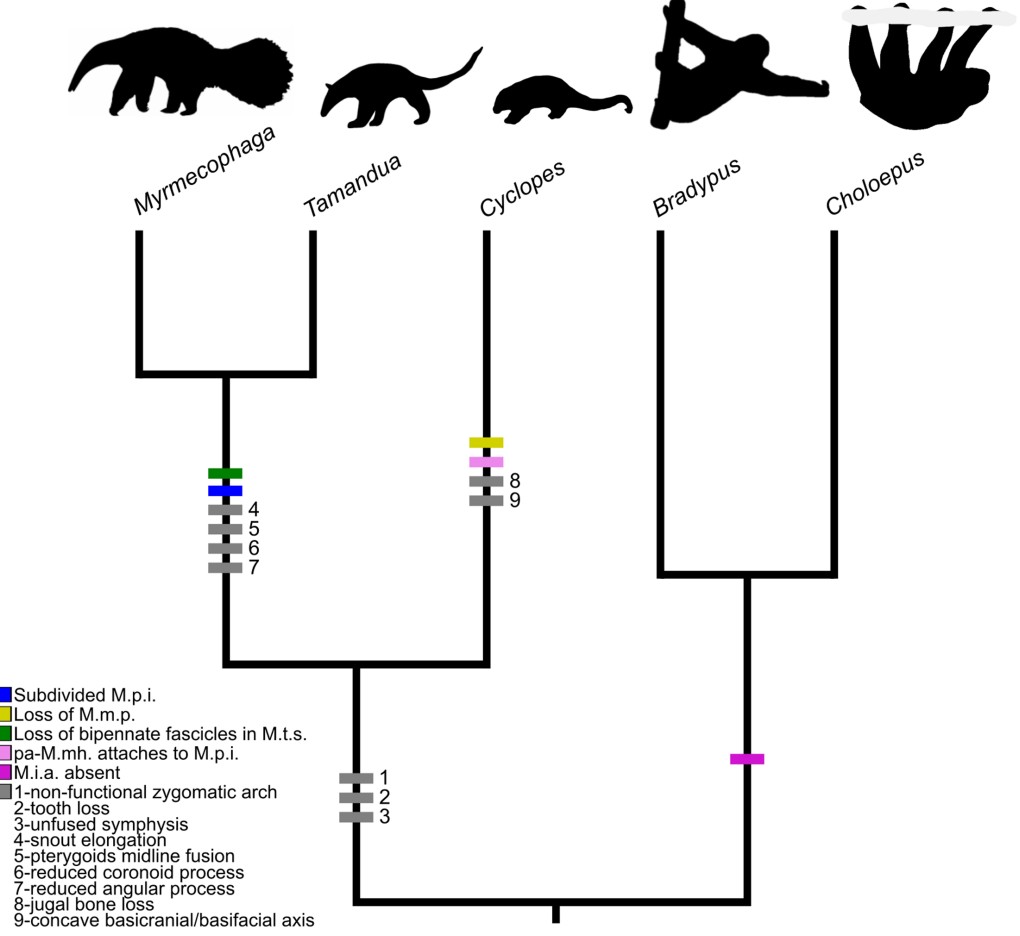

**Figure 10 Mapping of muscular and osteological discrete traits in simplified phylogeny of Pilosa.** Trait 1 refers to the absence of a maxilla-jugal-suqamosal functional unit providing a surface for muscular origins; extant Pilosa all lack completely ossified zygomatic arches, but sloths present strong ligaments connecting the jugal and the zygomatic process of the squamosal from which the *M. zygomaticomandibularis* and the *M. masseter profundus* arise (*Naples, 1985b*). Traits 2–11 are based on cranial synapomorphies, directly related to muscular origins/insertions, described in *Hirschfeld (1976)*, *Engelmann (1985)*, and *Gaudin & Branham (1998)*. The tree was obtained from timetreeoflife.org (*Kumar et al., 2017*) and divergence times were modified according to *Gibb et al. (2016)*. Silhouettes correspond to one species within the tip taxon.

These, and other morphological traits, are considered ancestral for Vermilingua (Fig. 10; *Hirschfeld, 1976*; *Patterson et al., 1992*). Reiss (*1997*; *2001*) also found differences between the head musculature of pygmy and myrmecophagid anteaters but overlooked those in the masticatory apparatus.

Our results reveal clear differences in the anatomy of the masticatory muscles of anteaters (Fig. 10). Contrary to myrmecophagids, the pygmy anteater shows a simple *M. pterygoideus internus* (M.p.i.) without subdivisions, a one-layered *M. masseter* (*superficialis*), and a relatively larger *M. temporalis superficialis* (M.t.s.) with a bipennate fascicular architecture (Fig. 10). Additionally, the posterior part of the *M. mylohyoideus pars anterior* (pa-M.mh.)

inserts on the ventromedial part of the *M. pterygoideus internus*, unlike in myrmecophagids (this study; *Naples, 1999*; *Endo et al., 2007*; *Endo et al., 2017*). Lastly, we show the existence of a two-part *M. buccinatorius* in the pygmy anteater, contradicting previous descriptions (*Naples, 1985a*; *Reiss, 1997*). These five traits are of potential systematic value but all were absent in previous comparative studies identifying phylogenetically polarised muscular traits (*Reiss, 1997*; *Reiss, 2001*).

The subdivision of the *M. pterygoideus internus* into two parts in myrmecophagids might be related to size, similar to the increase in the number of facial muscles in anteater species with longer rostra (*Naples, 1985a*). On the other hand, size differences between collared and giant anteaters does not affect the *M. pterygoideus internus* anatomy. The subdivision of this muscle might thus be a diagnostic trait within Vermilingua.

*Reiss (1997)* failed to identify a complex *M. masseter* (with deep and superficial muscles) in the Northern tamandua and the giant anteater. Our description of a two-unit *masseter* musculature in myrmecophagids supports the observations made by *Endo et al. (2007)* and *Endo et al. (2017)*, and resembles that of other mammalian groups (e.g., *Turnbull, 1970*; *Naples, 1985b*; *Endo et al., 1998*; *Cox & Jeffery, 2011*; *Sharp & Trusler, 2015*). A single-unit *masseter* musculature is therefore an autapomorphy of Cyclopedidae. In the latter taxon, the muscle is attached to the maxilla by a long tendon (Figs. 3A and 3B). In addition to the lack of an *M. masseter profundus* (M.m.p.), *C. didactylus* displays a bipartite *M. masseter superficialis* (pa-M.m.s. and pp-M.m.s.; Figs. 3A, 3C, and 3D), while it is composed of a single block in myrmecophagids (Figs. 6A, 6C, 8A and 8C). The pa-M.m.s. in *C. didactylus* is distinguishable from an M.m.p. because: (i) it presents a *pars* reflexa, typically found in the M.m.s. (e.g., *Sharp & Trusler, 2015*); (ii) it shares a single tendinous origin with the pp-M.m.s.; (iii) a two part M.m.s. with differently orientated muscle fascicles is described in other mammals (e.g., Fig. 3A, *Sharp & Trusler, 2015*; *Wille, 1954*).

The *temporalis* complex is also quite distinctive between cyclopedids and myrmecophagids, despite both families presenting deep and superficial muscles (*contra Reiss, 1997*). The *temporalis* complex is twice as large in cyclopedids compared to myrmecophagids (Table 2). Robust jaw adductor muscles represent an ancestral condition within xenarthrans (*Reiss, 2001*). Therefore, the presence of large *M. temporalis superficialis* and *profundus* in pygmy anteaters is in line with other plesiomorphic musculoskeletal traits previously described (*Hirschfeld, 1976*; *Engelmann, 1985*; *Reiss, 1997*). The bipennate fascicular arrangement of the *M. temporalis superficialis* in the pygmy anteater (Fig. S2B) is an ambiguous trait. While it is unique to pygmy anteaters within Vermilingua, fiber pennation is not described in the sloth sister-group (*Naples, 1985b*). Nevertheless, the loss of bipennate fascicles in the *M. temporalis superficialis* might be an autapomorphic trait of myrmecophagids, given that other mammals present either bipennate or multipennate fiber arrangements (*Woods & Howland, 1979*; *Taylor & Vinyard, 2009*; *Hautier, 2010*). Curiously, the *pars zygomatica* of the *M. temporalis superficialis* is relatively smaller in *C. didactylus* than in myrmecophagids (Table 2), suggesting that the posterior component of force of the *temporalis* complex is less important in pygmy anteaters.

In addition, to the differences listed above, we recognize, for the first time, the presence of an individualized *M. intermandibularis anterior* (M.i.a.) in the Vermilingua (Figs. 5B,

5B, and 9B). *Naples (1999)* considered this muscle to be a part of the *M. mylohyoideus* (M.mh.). We show that M.i.a. is attached to the ventrolateral margin of the anterior part of the lower jaws (Figs. 5B, 5B, and 9B), which contrasts with the insertion area of the M.mh. Furthermore, we confirm that the M.i.a. is made of transversally continuous fibers. The pa-M.mh. and pp-M.mh. comprise two bilaterally symmetric muscles that join along a midline axis (Fig. S2C). A similar condition is found in sloths (*Naples, 1986*), as well as in other mammals like moonrats (*Turnbull, 1970*), nectarivorous bats (*Wille, 1954*), and humans (*Gray, 1995*). *Turnbull (1970)* uses two criteria to assign a *M. digastricus pars anterior* to the M.mh.: (i) the presence of intertonguing connection at the midline, and (ii) the contiguity of the attachment on the mandible. None of these conditions were found in the anteater "pa-M.mh." (*sensu Naples, 1999*). Therefore, we propose to consider this muscle as the M.i.a. (*Diogo et al., 2008*). The pa-M.mh. (*sensu Naples, 1999*), the *M. transversus mandibularis* of rats (*Greene, 1935*), and the pa-M.mh. of tree-shrews (*Le Gros Clark, 1924*) are developmentally distinct from the M.mh. (*Diogo et al., 2008*). The muscle referred to by *Le Gros Clark (1924)*, *Greene (1935)*, and *Naples (1999)* is developmentally homologous with the sarcopterygian M.i.a. while the M.mh. is homologous to the *M. intermandibularis posterior* (*Diogo et al., 2008*). The M.i.a. muscle is mostly present in mammals with highly mobile mandibular symphysis, serving as a stabilizer (*Hiiemae & Houston, 1971*).

Overall, the results of our detailed descriptions and comparisons of the masticatory apparatus of anteaters provide several morphological traits that can be useful for systematics purposes. The previously unaccounted differences between the masticatory muscles of cyclopedids and myrmecophagids emphasize the level of morphological divergence acquired during the evolution of this clade with a highly specialized diet. We highlight the importance of soft-tissues as a source of diagnostic traits by combining conventional dissection with dice-CT (*Metscher, 2009*). Our results allow us to propose that a two part *masseter* musculature associated with a jugal bone and an unfused mandibular symphysis presenting an *M. intermandibularis anterior* are the plesiomorphic condition for Vermilingua. On the other hand, plesiomorphic architecture and relative size of the *temporalis* complex are impossible to predict, as these differ between extant sloth genera (*Naples, 1985b*) and data for armadillos (their xenarthran outgroup) are scarce and inconclusive (*Kuhlhorn, 1939 in Turnbull, 1970*).

## Mandibular mechanics

Regardless of the numerous differences discussed in the previous section, the masticatory apparatus of anteaters can be generally characterized by a set of adaptations to myrmecophagy like the complete tooth loss, the loss of masticatory capabilities (*Naples, 1999*), the reduction of masticatory muscles (*Reiss, 1997*; *Naples, 1999*; *Endo et al., 2007*; *Endo et al., 2017*), and the unfused mandibular symphysis (e.g., *Ferreira-Cardoso, Delsuc & Hautier, 2019*). The loss of chewing ability is well illustrated by the absence of the main mandibular abductor, the *M. digastricus* (e.g., *Turnbull, 1970*; *Hylander, Johnson & Crompton, 1987*; *Hylander, 2006*) in all dissected specimens.

The loss of a typical mandibular adduction/abduction cycle evolved with a new feeding strategy involving protrusion-retraction movements of an elongated sticky tongue (*Montgomery, 1983*; *Montgomery, 1985*). *Naples (1999)* associated this type of movement with the unfused mandibular symphysis in the giant anteater. The proposed model suggests that the loose symphysis allows for hemimandibular roll in order to increase the volume in the oral cavity (mouth opening) during tongue protrusion (*Naples, 1999*; Figs. 11A–11E). The medial roll of the dorsal margin of the mandibular body (mouth opening; Fig. 11D) is achieved by the contractions of the masseter complex and the and *M. temporalis superficialis*. The former contributes to the lateral roll of the angular process of the mandible, while the latter contributes to the medial roll of the coronoid process and additionally performs retraction movements (Fig. 11A–11C; *Naples, 1999*). Mouth closing (Fig. 11E) results from the lateral roll of the dorsal edge of the mandible, which is achieved by the contraction of the *M. pterygoideus internus* (Figs. 11B and 11C; *Naples, 1999*). The *M. temporalis profundus* also contributes to mandibular closing by medially rolling the ascending ramus (Figs. 11B and 11C; *Naples, 1999*). The contraction of the well-developed *M. intermandibularis anterior* (*M. mylohyoideus pars anterior sensu Naples, 1999*) additionally contributes to hemimandibular roll (*Naples, 1999*). This contraction medially rotates the ventral margin of the mandibular rami, causing the lateral roll (Fig. 11E) of their dorsal edges (mouth closing; *Naples, 1999*). Collared anteaters probably show similar mandibular mechanics as they show many anatomical similarities with giant anteaters (Figs. 11A–11E; *Endo et al., 2017*).

A biomechanical model of the masticatory apparatus of pygmy anteaters is yet to be proposed. On the one hand, cyclopedids and myrmecophagids present several muscular and osteological differences (see previous section of 'Discussion'). On the other hand, key similarities such as a large *M. intermandibularis anterior*, a reduced *masseter complex*, and an unfused mandibular symphysis suggest that both families share the same roll-dominated hemimandibular movements. Additionally, all anteaters present a typical mandibular innervation pattern composed of dorsal canaliculi that were putatively associated to the coordination between hemimandibular rolling and tongue protrusion in anteaters (*Ferreira-Cardoso, Delsuc & Hautier, 2019*).

We propose that food ingestion in pygmy anteaters happens through hemimandibular roll similar to that in myrmecophagids. However, the large coronoid process/*temporalis* musculature in pygmy anteaters suggest a relatively higher bite force magnitude (*Jones, 1997*; *Jaskolka, Eppley & Van Aalst, 2007*; *Nogueira, Peracchi & Monteiro, 2009*). The evolution of a large *temporalis* complex is associated with an increase in crushing force (e.g., *Jones, 1997*). Pygmy anteaters toothlessness and associated myrmecophagous diet suggest that relative muscular volumes are insufficient to characterize the masticatory mechanics, and that different mandibular mechanics may result from similar muscle proportions. The bipennation of the pygmy anteater *M. temporalis superficialis* indicates that this muscle is likely responsible for a majority of the force applied during mandibular movement (see 'Muscle-bone interactions'; *Avis, 1959*; *Amorim et al., 2008*). Therefore, the mediolateral roll is likely *temporalis*-led in pygmy anteaters.

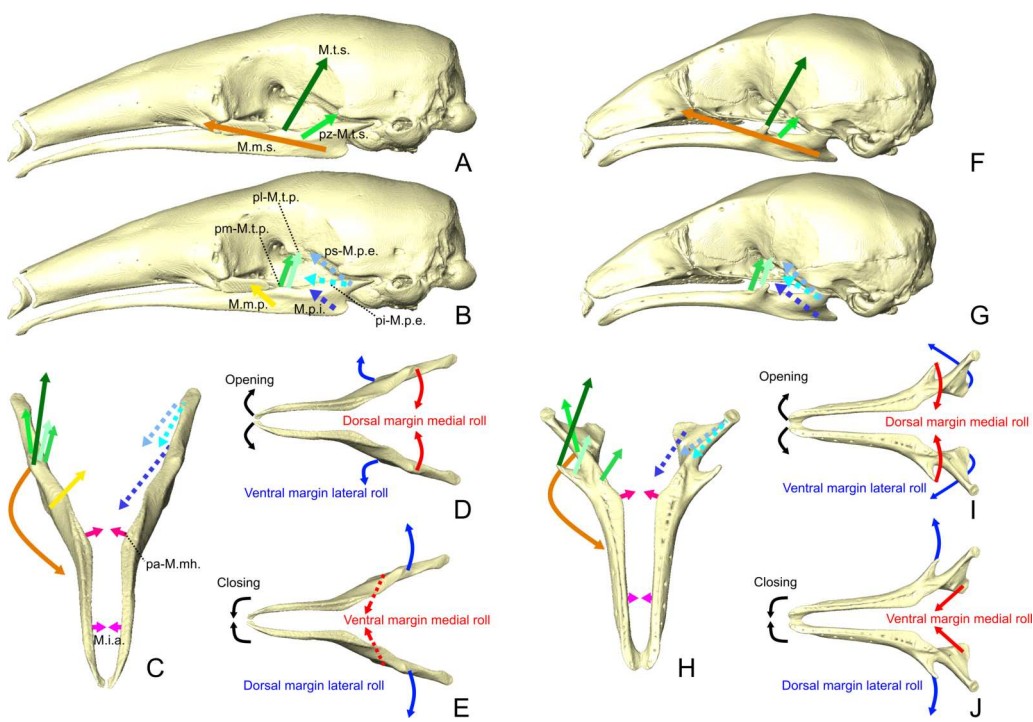

**Figure 11 Masticatory and intermandibular muscles lines of action and mandibular dynamics in *T. tetradactyla* (A-E) and *C. didactylus*(F-J).** Lateral views of the skull and mandible with the anteroventral pull directions of the superficial (A, F) and deep (B, G) muscles (A, D). Anterodorsal view of the mandibles with the mediolateral component of the lines of action (C, H). Schematic illustration of the mediolateral rotation mandibular movement (*Naples, 1999*) during mandibular opening (D, I) and closing (E, J). Lines of action color code corresponds to that use for the muscles. Dotted lines represent the lines of action of muscles completely or partially not visible in lateral view. Muscle abbreviations as in Table 1.

We argue that the lateral orientation of the coronoid process (Figs. 3B, 3A, 11F−11H) and the *M. temporalis superficialis* medial line of action in the pygmy anteater (Fig. 11H) are the basis for the *temporalis*-led medial roll of the dorsal margin of the mandibular body (mouth opening; Fig. 11I). This contrasts with mouth opening in myrmecophagids, in which the lateral roll of the angular process is putatively led by the *M. masseter superficialis* (larger relative contribution; Table 2). A similar *temporalis*-led hemimandibular roll is found in the tailless tenrec (*Tenrec ecaudatus*; *Oron & Crompton, 1985*), which presents a highly mobile mandibular symphysis (*Mills, 1966*) and lacks a *M. masseter profundus* (*Oron & Crompton, 1985*). These traits grant a high mediolateral mobility during mandibular adduction (*Oron & Crompton, 1985*). Interestingly, *masseter/temporalis* relative proportions in the tailless tenrec (*Turnbull, 1970*) are similar to those of the pygmy anteater, and so is the lateral orientation of their coronoid process (e.g., Figs. 11F−11H). The absence of a *M. masseter profundus* implies that the force vector of the *masseter* complex has a much reduced vertical component compared to other mammals (*Weijs, 1980*; *Gueldre & De Vree, 1990*; *Cox et al., 2012*). In addition, we propose that the contraction of the transverse fibers of the *M. mylohyoideus pars anterior* (*M. mylohyoideus pars media sensu Naples, 1999*) also

contributes to the medial roll of the dorsal margin of the ascending ramus (Fig. 11H), by applying a ventromedial force (Figs. 5C, 7C, and 9C). The contraction of the *M. pterygoideus internus*, *M. temporalis profundus pars medialis*, and *M. intermandibularis anterior* applies the medial force on the ventral margin of the mandibular ramus (Fig. 11H) during mouth closing (Fig. 11J). In opposition to myrmecophagids, the *pars reflexa* of the *M. masseter superficialis* in the pygmy anteater is relatively well-developed and wraps around the full length of the ventral margin of the ascending ramus (Fig. S2A). This might indicate that the *M. masseter superficialis* retains an elevator function (*Hylander, 2006*), and that it might facilitate the mouth closing, by adducting the mandibular rami (e.g., *Hiiemae, 1971*).

In sum, cyclopedids and myrmecophagids evolved similar mandibular movements (roll-dominated) despite the evolution of divergent skull shapes and sizes (*Reeve, 1940*). This functional adaptation of mandibular dynamics to tongue protrusion/retraction cycles in Vermilingua could represent a new example of many-to-one-mapping of form to function (*Wainwright et al., 2005*; *Strobbe et al., 2009*). Future functional comparisons between pygmy and myrmecophagid anteaters should include estimations of mechanical and bite force potential (e.g., *Cornette et al., 2012*), and a better characterization of the functional output of the jaw system using biomechanical models (e.g., *Cleuren, Aeris & De Vree, 1995*).

## Muscle-bone interactions

In the previous sections we discussed the differences between the *masseter* and *temporalis* muscle complexes between cyclopedids and myrmecophagids. The osteological divergence between these families partly reflects the *temporalis/masseter* trade-off that is key to understand the similar functional output of their masticatory apparatus. While the *masseter* musculature is reduced in cyclopedids, the *M. temporalis superficialis* is especially enlarged when compared to myrmecophagids (Table 2). These muscles' origin and insertion areas correspond to major osteological differences between pygmy and myrmecophagid anteaters (e.g., *Reiss, 1997*; *Ferreira-Cardoso, Delsuc & Hautier, 2019*). On the one hand, pygmy anteaters lack a jugal bone, which is the origin of the *M. masseter profundus* in myrmecophagids (this study; *Naples, 1999*; *Endo et al., 2017*) and other mammals (*Turnbull, 1970*; *Cox & Jeffery, 2011*; *Fabre et al., 2017*; *Ginot, Claude & Hautier, 2018*). On the other hand, the main surface for the insertion of the *M. temporalis superficialis*—the coronoid process—has almost vanished during the evolution of myrmecophagids. This observed covariation pattern between muscles and bones can be partly explained by the muscle-bone interactions that occur during embryonic development (*Cheverud, 1982*; *Hallgrímsson et al., 2007*; *Zelditch et al., 2008*). The absence/reduction of muscular contraction results in skeletal anomalies, including the loss or reduction of certain bones and cartilages (*Hall & Herring, 1990*; *Atchley & Hall, 1991*; *Rot-Nikcevic et al., 2006*).

The loss of the jugal bone in pygmy anteaters could be linked to the absence of a *M. masseter profundus* (M.m.p.). Similar conditions are present in the tailless tenrec (*Tenrec ecaudatus*; *Oron & Crompton, 1985*) and the Asian house shrew (*Suncus murinus*; *Fearnhead, Shute & Bellairs, 1955*). However, the absence of a jugal bone does not always imply the loss of the M.m.p. (e.g., *Crocidura russula*; *Cornette, Tresset & Herrel, 2015*).
Such cases may result from either early bone fusion (e.g., jugal + maxilla; *Tavares et al., 2017*), or the inactivation of genes that induce bone development (e.g., *Kist, Greally & Peters, 2007*). Additionally, the M.m.p. could either not differentiate during ontogeny or secondarily fuse with the *M. masseter superficialis*, as reported for other muscle complexes (e.g., *Diogo, 2018*). Most mammals (e.g., *Turnbull, 1970*; *Sharp & Trusler, 2015*), including sloths (*Naples, 1985b*) and myrmecophagid anteaters, present jugal bones and an M.m.p. Therefore, the complete loss of a functional zygomatic arch and the M.m.p. in the pygmy anteater offers a striking example of developmental integration linked to muscle-bone interaction, as well as an empirical evidence of modularity within the masticatory apparatus.

The reduction of both the coronoid process of the mandible and the *M. temporalis superficialis* in myrmecophagid anteaters also represents a classical example of structural covariance. Although the initiation of the coronoid development is an intrinsic process to the mandibular ossification, its growth is dependent on mechanical loading applied by the *temporalis* musculature (*Avis, 1959*; *Amorim et al., 2008*; *Anthwal, Peters & Tucker, 2015*). *Anthwal, Peters & Tucker (2015)* showed that reduced *temporalis* musculature correlated with weakly-developed coronoid processes in mice. The contrast between the large *M. temporalis superficialis* (42.2%) and prominent coronoid process in pygmy anteaters, and the much smaller muscle (13.5%–20.9%) and almost nonexistent process in myrmecophagids is a good example of muscle-induced coronoid development in non-model organisms. In addition to size, bipennate muscles in pygmy anteaters generate relatively larger forces (increased physiological cross-section areas) than unipennate ones in myrmecophagids (*Turnbull, 1970*; *Gans & De Vree, 1987*; *Hylander, 2006*), which further indicates a decrease of medially rotating forces applied on the coronoid process in the myrmecophagids.

While the examples discussed above are a good illustration of contrasting morphofunctional patterns between cyclopedids and myrmecophagids, the loss of the zygomatic arch represents a common developmental trend. All anteaters putatively lack a *M. zygomaticomandibularis* muscle (*Edgeworth, 1923*; *Naples, 1999*). This muscle originates from the zygomatic arch in sloths (*Naples, 1985b*) and other mammals (*Turnbull, 1970*; *Cox & Jeffery, 2011*; *Sharp & Trusler, 2015*). Although the absence of muscular contraction by the *M. zygomaticomandibularis* muscle could provide a developmental explanation for the loss of the zygomatic arch (e.g., *Hall & Herring, 1990*; *Herring, 1993*), we argue that the this muscle was not completely lost in anteaters. We found a *pars zygomatica* of the *M. temporalis superficialis* in the three species of anteaters, especially well-separated in the pygmy anteater. *Naples (1999)* homologized this muscle in *M. tridactyla* with a homonymous structure in the two-toed sloth (*Choloepus* sp.). However, the *pars zygomatica* of the *M. temporalis superficialis* of anteaters inserts along the lateral part of the mandibular notch (Figs. 1B, 3, 6 and 8), instead of the anterior edge of the coronoid process as in the two-toed sloth (1985b). Therefore, we dispute the homology implied by *Naples (1999)*, and propose that the *pars zygomatica* of the *M. temporalis superficialis (pz-M.t.s.)* of anteaters might correspond to a *M. zygomaticomandibularis pars posterior (pp-M.zm.)*. The origin and insertion of *the pz-M.t.s* are similar to those of the pp-M.zm. described in rodents (e.g., *Cox & Jeffery, 2011*; *Fabre et al., 2017*; *Ginot, Claude & Hautier, 2018*). Previous descriptions of

this muscle as a part of the *M. temporalis superficialis* (*Naples, 1985b*; *Naples, 1999*) can be justified by the common developmental origin of the two muscles (e.g., *Edgeworth, 1914*). Previous studies described the *M. zygomaticomandibularis* as not separable from either the *masseter* or the *temporalis* complexes in some carnivores, ungulates, bats, and marsupials (*Druzinsky, Doherty & De Vree, 2011* and references therein). Nevertheless, we propose that the designation of *pars zygomatica* of the *M. temporalis superficialis* should be used until further embryological evidence is available in anteaters.

## Dietary *versus* functional convergence

Myrmecophagy is a textbook example of evolutionary convergence linked to dietary adaptation (*McGhee, 2011*). However, a comprehensive comparative study of the masticatory apparatus of all ant- and termite-eating placental lineages is yet to be undertaken. *Reiss (2001)* took a first step in this direction, but this study included pangolins, anteaters, and respective sister taxa only. Furthermore, *Reiss (2001)* focused on the systematic implications of convergence (i.e., homoplasy), with morphofunctional considerations focusing mostly on tongue musculature.

Pangolins are a well-known example of ecological and morphological convergence with anteaters (*Rose et al., 2005*; *McGhee, 2011*). While several studies have been dedicated to the head musculature of pangolins (e.g., *Macalister, 1875*; *Windle & Parsons, 1899*; *Edgeworth, 1923*; *Imai, 1978*; *Endo et al., 1998*), quantitative elements (volume or mass ratios) and functional interpretations are almost nonexistent. Despite this lack of information, both groups present evident muscular differences. First, the *masseter* complex appears to be more complex than in anteaters, with three parts described in *Manis javanica* (Sunda pangolin; *Endo et al., 1998*; Fig. 7). *Windle & Parsons (1899)* reported that the *masseter* takes its origin on a "fibrous zygoma". Differences in fiber orientation are not provided (*Endo et al., 1998*), although the most anterior bundle appears to be the most vertically oriented (Fig. 7; *Endo et al., 1998*). This might suggest the existence of a *M. masseter superficialis* with two layers, as in *C. didactylus*, with a small *M. masseter profundus* (M.m.p.) anteriorly. However, this cannot be confirmed based on the existing bibliography. *Edgeworth (1923)* describes the *M. masseter* as arising from the "lower margin of the zygomatic portion of the superior maxilla", while a more oblique muscle arises from the medial surface of the zygomatic arch ("*M. zygomaticomandibularis*", M.zm.). The figures associated with *Edgeworth*'s (*1923*; Figs. 59 and 60) study suggest that the described M.zm. could also correspond to an M.m.p. (Table 4). Considering the available information, establishing clear homologies with the *masseter* complex of anteaters is not possible. Nonetheless, if pangolins have a "true" M.zm., this muscle is not homologous to a putative M.zm. in anteaters, given the much more posterior origin and insertion of the latter (see 'Muscle-bone interactions').

The *pterygoideus* complex of pangolins is composed of a *M. pterygoideus externus* (M.p.e.), and a *M. pterygotympanicus* (absent in anteaters; *Edgeworth, 1923*; *Endo et al., 1998*). *Edgeworth (1923)* described a *M. pterygoideus internus* (M.p.i.) that is atrophied during development, while *Yeh (1984)* and *Endo et al. (1998)* reported its absence in adult pangolins. As for the M.p.e., *Endo et al. (1998)* suggested the presence of two muscle bundles, probably corresponding to the pi-M.p.e. and the ps-M.p.e. of anteaters and other

**Table 4  Proposed homologies for muscles previously described in pangolins (*Edgeworth, 1923*) and aardvarks (*Edgeworth, 1924*; *Sonntag, 1925*; *Frick, 1951*).**

| Edgeworth (1923) | Edgeworth (1924) and Sonntag (1925) | Frick (1951) | This study |
|---|---|---|---|
| digastricus anterior | digastricus posterior | biventer | **M.di.** |
| intermandibularis anterior | digastricus anterior | mylohyoideus | **pa-M.mh.** |
| intermandibularis posterior | digastricus anterior | mylohyo ideus | **pp-M.mh.** |
| geniohyoideus | intermandibularis | geniohyoideus | **M.gh.** |
| zygomaticomandibularis* | zygomaticomandibularis | masseter (schicht 3+4) | **M.zm.** |
| – | temporalis ant. portion** | pars medialis - temporalis | **M.t.s.+pz-M.t.s.** |
| – | temporalis post. portion** | pars posterior - temporalis | **Absent** |
| – | temporalis inner part*** | pars orbitalis - temporalis | **M.t.p.** |

**Notes.**

*Sonntag (1925)* did not describe the parts of the *Mm. temporalis*.

*this could correspond to a *M. masseter profundus*

**included in the "outer part"

***not explicit in the text

M.di, *M. digastricus*; pa-M.mh., *M. mylohyoideus pars anterior*; pp-M.mh., *M. mylohyoideus pars posterior*; M.gh., *M. geniohyoideus*; M.zm., *M. zygomaticomandibularis*; M.t.s., *M. temporalis superficialis*; pz-M.t.s., *M. temporalis superficialis pars zygomatica*; M.t.p., *M. temporalis profundus*.

mammals (e.g., *Turnbull, 1970*). While the ps-M.p.e. arises from the parietal bone in anteaters, both parts arise from the pterygoid in pangolins (*Endo et al., 1998*).

Unlike anteaters, pangolins appear to present an *M. digastricus* (M.di.; *Windle & Parsons, 1899*; *Edgeworth, 1923*; *Endo et al., 1998*; Table 4). It is, nevertheless, less developed than in other mammals (e.g., *Turnbull, 1970*).

In pangolins, both *M. intermandibularis anterior* (M.i.a.) and *posterior* (M.i.p.) have been described (*Edgeworth, 1923*). However, both present a medial raphe, which suggests that these elements are homologous to the *M. mylohyoideus pars anterior* (pa-M.mh.) and *posterior* (pa-M.mh.) of anteaters (Table 4). The attachment of the latter to the pterygoid bone (*Edgeworth, 1923*) is similar to the condition in *T. tetradactyla* (Fig. 7). *Endo et al. (1998)* described a single M.mh. in *M. javanica*, but no M.i.a. The available information suggests the absence of a true M.i.a. (*sensu Diogo et al., 2008*) in pangolins, which is congruent with the presence of a fused symphysis.

The divergence between the musculature of anteaters and pangolins reflects the distinct evolutionary histories of these two placental groups (*Meredith et al., 2011*). Furthermore, key aspects of the pangolin masticatory apparatus like the absence of an *M. intermandibularis anterior*, the presence of an *M. digastricus*, and the fused mandibular symphysis suggest the evolution of completely different mechanics from the hemimandibular roll in anteaters. Different nomenclatures, heterogeneous levels of details, and limited illustrations in previous studies make a unifying study of the head musculature of pangolins a prerequisite for further interpretations.

The aardvark (*Orycteropus afer*) head musculature is substantially better described than that of pangolins. The head muscles of the aardvark have been the subject of several studies (*Humphry, 1868*; *Galton, 1869*; *Windle & Parsons, 1899*; *Bender, 1909*; *Edgeworth, 1924*; *Sonntag, 1925*; *Frick, 1951*). Similar to other myrmecophagous species, the aardvark

also possesses an elongated and specialized tongue (*Goździewska-Harłajczuk, Klećkowska-Nawrot & Barszcz, 2018*), an elongated snout, and a regressed dentition lacking enamel. However, the aardvark presents a developed ascending ramus of the mandible, with prominent coronoid and condylar processes, a broad masseteric fossa (e.g., *Edgeworth, 1924*; *Ferreira-Cardoso, Delsuc & Hautier, 2019*), and is able to chew (*Patterson, 1975*).

The *masseter* complex of *O. afer* is composed of multilayered *M. masseter superficialis* (M.m.s.) and *profundus* (M.m.p.; *Frick, 1951*). Similar to anteaters, the M.m.s. is the largest (*Frick, 1951*). According to *Frick (1951)* it is divided into three layers, the anterior two originating from the zygomatic process, while the most posterior originates from the jugal. The M.m.s. presents a large *pars reflexa* posterodorsally (*Edgeworth, 1924*; *Sonntag, 1925*; *Frick, 1951*). The M.m.p. of *O. afer* is divided into two sublayers (*Frick, 1951*). The architecture of the *masseter* complex of *O. afer* appears to be more complex than that of both anteaters and pangolins.

Unlike anteaters, *O. afer* presents a *M. zygomaticomandibularis* (M.zm.) that inserts on the lateral surface of the mandible dorsally to the *masseter* complex. This muscle has vertically oriented fibers and originates along the posteroventral part of the jugal and the anterior part of the zygomatic process of the squamosal (*Edgeworth, 1924*; *Sonntag, 1925*; *Frick, 1951*). It is probably not related to the *M. temporalis superficialis pars zygomatica* of anteaters, as its insertion on the mandible is much more anteroventral and its origin further apart from the temporal fossa (*Edgeworth, 1924*).

The *temporalis* complex of *O. afer* differs from that of anteaters as it extends dorsally and posteriorly into the cranial vault and fully covers the coronoid process (*Edgeworth, 1924*; *Sonntag, 1925*). This complex is divided into three parts (*Frick, 1951*). The posterior part is the largest and might be homologous to the *M. temporalis superficialis* (M.t.s.) and its *pars zygomatica* (pz-M.t.s.) in anteaters (Table 4). In contrast to anteaters, the origin of this muscle stretches posterodorsally into the parietal (*Edgeworth, 1924*; *Sonntag, 1925*; *Frick, 1951*). The smaller medial part of the *masseter* complex of *O. afer* appears to be absent in anteaters. Firstly, fiber separation of a medial portion of the M.t.s. was absent in vermilinguans. Secondly, the medial part of the M.t.s. of *O. afer* arises from the most anterior part of the parietal and the posterior part of the postorbital process (*Frick, 1951*), which is not developed in anteaters. The deepest part of the *M. temporalis* of *O. afer* (*Frick, 1951*) corresponds to the *M. temporalis profundus* of anteaters (Table 4).

The *pterygoideus* complex in *O. afer* presents some differences when compared to that of anteaters (*Edgeworth, 1924*; *Sonntag, 1925*; *Frick, 1951*). Its *M. pterygoideus internus* presents three parts, while only two were identified in *M. tridactyla* and *T. tetradactyla*, and a single one in *C. didactylus*. The *M. pterygoideus externus* is similar to that of anteaters, being separated into superior and inferior heads arising from the alisphenoid (*Frick, 1951*), although neither *Edgeworth (1924)* nor *Sonntag (1925)* refer to such division. As in pangolins, the aardvark presents a *pterygotympanicus* that originates on the ectotympanic and displays a tendinous connection to the *tensor veli palatini*, on the palate (*Edgeworth, 1924*; *Sonntag, 1925*).

*Edgeworth (1924)* described a longitudinally oriented "*intermandibularis*" with bifid tendinous insertions on the ventrolateral surface of the mandibles of *O. afer*. *Sonntag*

*(1925)* suggested that this muscle might correspond to the *M. mylohyoideus* (M.mh.). We propose that the aardvark's "*intermandibularis*" is homologous to the *M. geniohyoideus* (M.gh.) of anteaters (Table 4), as both have longitudinally oriented fibres, present an anterior bifurcation of the muscular fibers, and take their origin on the ceratohyal and basihyal (*Edgeworth, 1924*; *Sonntag, 1925*; *Frick, 1951*). Such traits are present in other mammals such as sloths (*Naples, 1986*), dogs (*Evans & De Lahunta, 2013*; e.g., Figs. 6–22), and humans (*Drake et al., 2015*). This homology appears to agree with *Frick (1951)*, despite this author's introduction of a new term, "*intermandibularis profundus*", and contradictory illustration showing the bifurcate tendon associated with the M.mh. (Abb. 8; *Frick, 1951*). *Edgeworth (1924)* and *Sonntag (1925)* suggested that a transversely oriented muscle attaching medially to the posterior half of the aardvark mandible to be a "*M. digastricus anterior*". This muscle presents a variable median raphe (*Edgeworth, 1924*; *Frick, 1951*), such as the *mylohyoideus* complex in anteaters and other mammals (*Edgeworth, 1914*; *Saban, 1968*; *Turnbull, 1970*), and it does not connect to the *M. digastricus pars posterior* (*Edgeworth, 1924*; *Sonntag, 1925*). Therefore, we agree with *Frick (1951)* interpretation that the "*M. digastricus anterior*" (*sensu Edgeworth, 1924*; *Sonntag, 1925*) of the aardvark is homologous to the M.mh. (Table 4). It probably corresponds to the *M. mylohyoideus pars anterior* of anteaters, given its insertion area. The *M. mylohyoideus pars posterior* is also possibly present in *O. afer Frick (1951)*, but this remains to be confirmed.

In *O. afer*, both *pars anterior* and *posterior* of the *M. digastricus* (M.di.) appear to be present (*Humphry, 1868*; *Frick, 1951*), while both are absent in anteaters. The presence of the M.di. suggests that mandibular depression in *O. afer* happens as in other mammals (e.g., *Turnbull, 1970*; *Hylander, Johnson & Crompton, 1987*), in contrast with anteaters.

In sum, the aardvark masticatory musculature is much more similar to those of non-myrmecophagous placentals than to that of anteaters. This is not surprising, given the level of divergence between these two myrmecophagous lineages (*Meredith et al., 2011*), their different states of tooth reduction (*Meredith et al., 2009*), and their obvious differences in skull morphology (*Davit-Béal, Tucker & Sire, 2009*). Furthermore, the articular condyle is well dorsal to the tooth row in aardvarks, which generally increases the momentum of force applied to food items (*Greaves, 2012*). Despite being myrmecophagous, the aardvark actively chews in order to obtain water from cucumber-like plants (*Patterson, 1975*). The large *Mm. temporalis* and *masseter* (*Edgeworth, 1924*; *Sonntag, 1925*; *Frick, 1951*), the presence of an ossified zygomatic arch, and the fused mandibular symphysis imply significant functional differences from the anteater hemimandibular roll model. The masticatory apparatus in aardvarks does not appear to have undergone a morphofunctional shift during the evolution of myrmecophagy. Thus, aardvarks, pangolins, and anteaters constitute three examples of convergent dietary specialization with different mastication mechanics. This suggests that the plasticity of the mammalian masticatory apparatus played a key role in the convergent evolution of rapid ingestion of small-sized food items in myologically divergent lineages.

## CONCLUSION

Here we describe the masticatory, facial-masticatory, and intermandibular muscles of the three extant anteater genera. While collared and giant anteaters show very similar morphologies, the masticatory apparatus of the pygmy anteater exhibits marked differences. These include important discrete morphological traits with systematic value. We also propose that muscle-bone interactions play a major role in the morphological and functional differentiation between the two anteater lineages. Our proposed mastication model for the pygmy anteater suggests that tongue protrusion-retraction movements co-evolved with hemimandibular roll in both cyclopedids and myrmecophagids. This provides a fine example of many-to-one mapping (*Wainwright et al., 2005*; *Strobbe et al., 2009*), disagreeing with previous interpretations of similarity between the masticatory apparatus of the two anteater families (*Reiss, 1997*). Further comparison with available data from the literature shows that the biomechanics of anteaters may well differ from that of pangolins despite their ecological convergence (*McGhee, 2011*). Further studies will be needed to assess biomechanical differences, notably the magnitude of forces applied to the hemimandibular roll (e.g., *Oron & Crompton, 1985*). Such data will allow to precisely characterize mandibular movement in anteaters and pangolins and further explore the biomechanics of their masticatory apparatus in the context of convergent evolution towards myrmecophagy.

## ACKNOWLEDGEMENTS

We thank the three anonymous reviewers for their comments and suggestions. We acknowledge François Catzeflis (Institut des Sciences de l'Evolution de Montpellier) for providing access to the specimens used in this study. We would also like to thank Anthony Herrel and Kévin Le Verger for facilitating access to the specimens (Muséum National d'Histoire Naturelle de Paris). We thank Renaud Lebrun for his assistance with the μ-CT scanner, as well as the MRI platform. We would additionally like to thank Mélanie Debiais-Thibaud (Institut des Sciences de l'Evolution de Montpellier) and Samuel Ginot (Institut de Génomique Fonctionnelle de Lyon) for discussions about staining protocols, and Marie-Ka Tilak (Institut des Sciences de l'Evolution de Montpellier) for her help in the molecular biology laboratory. This is contribution ISEM 2020-188 of the Institut des Sciences de l'Evolution de Montpellier.

### Funding

Sérgio Ferreira-Cardoso, Lionel Hautier and Frédéric Delsuc were supported by a European Research Council (ERC) consolidator grant (ConvergeAnt project #683257). Lionel Hautier and Frédéric Delsuc were supported by Centre National de la Recherche Scientifique (CNRS). This work was supported by "Investissements d'Avenir" grants managed by Agence Nationale de la Recherche Labex CEBA (ANR-10-LABX-25-01), Labex CEMEB

(ANR-10-LABX-0004), and Labex NUMEV (ANR-10-LABX-0020). The MRI platform member of the national infrastructure France-BioImaging is supported by the French National Research Agency (ANR-10-INBS-04, 'Investments for the future'). The JAGUARS collection is supported through a FEDER/ERDF grant attributed to Kwata NGO, funded by the European Union, the Collectivité Territoriale de Guyane, and the DEAL Guyane. The funders had no role in study design, data collection and analysis, decision to publish, or preparation of the manuscript.

## Grant Disclosures

The following grant information was disclosed by the authors:
European Research Council (ERC): #683257.
Centre National de la Recherche Scientifique (CNRS).
Agence Nationale de la Recherche Labex CEBA: ANR-10-LABX-25-01.
Labex CEMEB: ANR-10-LABX-0004.
Labex NUMEV: ANR-10-LABX-0020.
French National Research Agency: ANR-10-INBS-04.
European Union.
Collectivité Territoriale de Guyane.
DEAL Guyane.

## Competing Interests

Benoît de Thoisy is the director and a scientific advisor at Kwata NGO. The remaining authors declare that they have no competing interests.

## Author Contributions

- Sérgio Ferreira-Cardoso conceived and designed the experiments, performed the experiments, analyzed the data, prepared figures and/or tables, authored or reviewed drafts of the paper, and approved the final draft.
- Pierre-Henri Fabre and Lionel Hautier conceived and designed the experiments, performed the experiments, authored or reviewed drafts of the paper, and approved the final draft.
- Benoit de Thoisy performed the experiments, authored or reviewed drafts of the paper, and approved the final draft.
- Frédéric Delsuc conceived and designed the experiments, authored or reviewed drafts of the paper, and approved the final draft.

## Data Availability

3D surfaces of the muscle reconstructions are available at Ferreira-Cardoso S., Fabre P.-H., de Thoisy B., Delsuc F., Hautier L., 2020. 3D models related to the publication: "Comparative masticatory myology in anteaters and its implications for interpreting morphological convergence in myrmecophagous placentals". MorphoMuseuM 6(2)-e114. doi: 10.18563/journal.m3.114

- Cyclopes didactylus M1571_JAG, doi: 10.18563/m3.sf.522

- Tamandua tetradactyla M3075_JAG, doi: 10.18563/m3.sf.524
- Myrmecophaga tridactyla M3023_JAG, doi: 10.18563/m3.sf.523

## Supplemental Information

Supplemental information for this article can be found online at http://dx.doi.org/10.7717/peerj.9690#supplemental-information.

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
