# Peer review of "Comparative masticatory myology in anteaters and its implications for interpreting morphological convergence in myrmecophagous placentals"

_PeerJ, doi:10.7717/peerj.9690_

## Round 0.1 · original submission · Minor Revisions

I have received three reviews of your paper. All of them concur in that it contains sound and interesting data. However, the reviewers have pointed out some inconsistencies and lack of coherence in the muscle nomenclature that need your attention. They also support including extensive descriptions as possible to facilitates future comparative studies. Please pay full attention to all their comments.

Reviewer 1 ·

Basic reporting

The language throughout is professional and generally clear. The level of written English is acceptable for publication.

The introduction was very expansive, covering a broad literature base relating to their focal taxa that was highly relevant and strong in scope. Similarly, the discussion well contextualizes their findings and clearly links these data to a wider body of existing work.

Results – particularly the comparative anatomical descriptions of each muscle (lines 228-726) are extremely thorough and do an excellent job of describing the anatomy of these taxa.
To this same end, the authors include a high number of figures, which are of excellent quality and well labelled/described.
PDF proof of 3D data uploaded to an accessible repository (MorphoMuseum) was provided in supplementary materials.

Experimental design

The study is appropriately designed to answer their question, which is novel, interesting, and well within the scope of the journal. The question at hand – though largely exploratory – addresses a clear and obvious knowledge gap regarding the anatomy of these enigmatic creatures, and they detail their findings in a manner well-fitting of such a study.

Technical components of the study design – i.e., the details of their staining, scanning, and reconstruction methology – is in-keeping with best practices of the field at this time. The methods as described are replicable to any scientist with a basic understanding of the technique at hand.

Validity of the findings

The findings of this paper are restricted to a largely descriptive overview of the anatomy, though some quantitative data in the form of muscle volumes are provided. Though the absolute volumes are likely reduced from their physiological true nature due to storage/preparation decisions (the combined effects of ethanol and iodine), the relative values between muscles intra-individually and especially between taxa inter-specifically are valuable. The broader benefit of their findings to comparative anatomists interested in the manifestation of form-function masticatory relationships within myrmecophagists is clear and though the audience scope is therefore relatively limited, the findings appear valid and worthy of dissemination. I have an issue with one particular finding (detailed below) where I believe data may either be oversold or misinterpreted, and would encourage the authors to be slightly more circumspect and less absolute with their discussion of this observation. Nonetheless, my overall impressions of the results are positive.

Additional comments

1. Small clarification needed to introduction, changing text discussing the jaw adductors from the ‘pterygoid’ to the ‘medial pterygoid’ to avoid conflating the abductive/translating lateral pterygoid as an adductor.

2. In results, lines 216-218, the authors note that their volumes were smaller than expected and suggest this was due to shrinkage associated with iodine staining. Though minor, this observation should cite a recent work by Hedrick et al. “Assessing Soft-Tissue Shrinkage Estimates in Museum Specimens Imaged With Diffusible Iodine-Based Contrast-Enhanced Computed Tomography (diceCT)” on this topic. It should also be made more explicitly clear in the methods whether the specimen was returned to ethanol for any significant period of time between dissection and staining, as if so, the dehydrating effects of additional ethanol exposure might also have contributed to this volumetric change.

3.The authors note an absence of the deep masseter (masseter profundus) within C. didactylus, which they illustrate in Figure 3a and note in lines 728-730. However, it would appear, at least from the figures and anatomical descriptions provided, that this observation is relatively semantic. The masseter superficialis within this taxon is noted to be bipartite (Figs 3C and 3D, line 734) and it would appear visually that the pars anterior is relatively distinctive in fascicular orientation with profundus fascicle more vertically oriented and superficialis fascicles more oblique. Thus, I would suggest rephrasing this observation to less definitively describe this muscle as ‘absent’ and rather note that the masseter superficialis and profundus are highly interdigitated, resulting in a single communal muscle volume that could not be differentiated within the scan. If the authors closer inspection of their data continue to support the notion of this muscle’s absence, a clearer justification for how this determination was made is necessary in the results.
(Anecdotally, this is a phenomenon I have observed in certain nectarivorous bats and this leads me to wonder whether extreme interdigitation of these two muscle portions might be more prevalent in non-masticating taxa (speculation beyond the realm of this manuscript, but perhaps of interest to the authors to consider).

4. The identification of a defined and distinct intermandibularis anterior is very interesting, and the authors make a compelling case to recognize this muscular portion as an independent structure, but for clarity I suggest referencing Figure 5B in the appropriate portion of their results (somewhere between lines 767 and 787) as this figure visually encapsulates many points which they describe extensively in text.

·

Basic reporting

This is all very good. I have provided minor edits on text in the attached document.

Experimental design

I am very glad to see unilateral dissection and unilateral digital dissection on the same specimen

Validity of the findings

Generally very good; section on muscle function is perhaps a little bit of a stretch, but creates testable hypotheses

Additional comments

Below, I have provided line-by-line feedback on minor linguistic edits and a few technical questions/comments. I have waived anonymity, and if you'd like to contact me to discuss further (especially the question I raise about the digastric!), you are welcome to.

Reviewer 3 ·

Basic reporting

See my uploaded review

Experimental design

See my uploaded review

Validity of the findings

See my uploaded review

Additional comments

See my uploaded review

Annotated reviews are not available for download in order to protect the identity of reviewers who chose to remain anonymous.

---

## Round 0.2 · accepted · Accept

Thank you very much for your careful attention to all reviewer suggestions. We are ready to go ahead.

Reviewer 1 ·

Basic reporting

This article satisfies requirements for basic reporting.

Experimental design

The design of this study is appropriate, relevant, and well applied.

Validity of the findings

The findings remain valid and conclusions are well supported by data.

Additional comments

I thank the authors for addressing the small number of comments I raised in my earlier review. These clarifications satisfy my suggestions, and I believe that the paper as a whole is well worthy of publication in its current form.

·

Basic reporting

No comment

Experimental design

No comment

Validity of the findings

No comment

Additional comments

I am satisfied with the changes to this manuscript.

Reviewer 3 ·

Basic reporting

The revised manuscript is now acceptable.

Experimental design

The revised manuscript is now acceptable.

Validity of the findings

The revised manuscript is now acceptable.

Additional comments

Thank you for addressing my major and minor concerns. I now have a better understanding of the jaw movements of ant-eating species and the relative terminology for the movements in each plane. I also appreciate the use of abbreviations in the Results section and more focus to the Discussion section.